# Cre/lox-assisted non-invasive in vivo tracking of specific cell populations by positron emission tomography

Martin Thunemann[1,8], Barbara F. Schörg[2], Susanne Feil[1], Yun Lin[2], Jakob Voelkl[3], Matthias Golla[1], Angelos Vachaviolos[1], Ursula Kohlhofer[4], Leticia Quintanilla-Martinez[4], Marcus Olbrich[5], Walter Ehrlichmann[2], Gerald Reischl[2], Christoph M. Griessinger[2], Harald F. Langer[5], Meinrad Gawaz[5], Florian Lang [3], Michael Schäfers[6], Manfred Kneilling[2,7], Bernd J. Pichler[2] & Robert Feil[1]

Many pathophysiological processes are associated with proliferation, migration or death of distinct cell populations. Monitoring specific cell types and their progeny in a non-invasive, longitudinal and quantitative manner is still challenging. Here we show a novel cell-tracking system that combines Cre/lox-assisted cell fate mapping with a thymidine kinase (sr39tk) reporter gene for cell detection by positron emission tomography (PET). We generate Rosa26-mT/sr39tk PET reporter mice and induce sr39tk expression in platelets, T lymphocytes or cardiomyocytes. As proof of concept, we demonstrate that our mouse model permits longitudinal PET imaging and quantification of T-cell homing during inflammation and cardiomyocyte viability after myocardial infarction. Moreover, Rosa26-mT/sr39tk mice are useful for whole-body characterization of transgenic Cre mice and to detect previously unknown Cre activity. We anticipate that the Cre-switchable PET reporter mice will be broadly applicable for non-invasive long-term tracking of selected cell populations in vivo.

[1] Interfakultäres Institut für Biochemie, University of Tübingen, 72076 Tübingen, Germany. [2] Department of Preclinical Imaging and Radiopharmacy, Werner Siemens Imaging Center, University of Tübingen, 72076 Tübingen, Germany. [3] Physiologisches Institut l, University of Tübingen, 72076 Tübingen, Germany. [4] Institute of Pathology and Neuropathology, University of Tübingen, and Comprehensive Cancer Center, University Hospital, 72076 Tübingen, Germany. [5] Department of Cardiovascular Medicine, University Hospital, University of Tübingen, 72076 Tübingen, Germany. [6] Department of Nuclear Medicine, University Hospital, European Institute for Molecular Imaging & EXC 1003 Cells-in-Motion Cluster of Excellence, University of Münster, 48149 Münster, Germany. [7] Department of Dermatology, University Hospital, University of Tübingen, 72076 Tübingen, Germany. [8] Present address: Department of Radiology, University of California San Diego, La Jolla, CA, USA. Correspondence and requests for materials should be addressed to R.F. (email: robert.feil@uni-tuebingen.de)

Tracking cells non-invasively in vivo by molecular imaging allows the observation of cell behavior in health and disease[1]. In addition to its importance for basic research, cell tracking has many potential applications in regenerative and individualized medicine and it facilitates the development of new diagnostic tools and therapeutic strategies[2–5]. Numerous imaging techniques are used to visualize cells in vivo, including ultrasound, optical imaging, magnetic resonance imaging (MRI) and positron emission tomography (PET). These methods require conceptually different labeling and detection strategies that each have inherent advantages and disadvantages. Direct cell labeling makes use of radioactive, fluorescent or paramagnetic compounds, which are, however, eventually washed out and get diluted. Thus, longitudinal and quantitative monitoring of cells becomes challenging. In contrast, strategies based on stable expression of a chromosomally integrated reporter transgene permit long-term labeling of cells and their progeny[1].

The Cre/lox recombination system has emerged as a powerful tool to generate time- and tissue-specific mouse mutants[6, 7]. In addition, this technology can be used to genetically label specific cell populations to map their fate during development[8] or in adult mice in the context of physiological or pathophysiological processes[9]. For genetically inducible fate mapping, cell type-specific expression of the tamoxifen-inducible CreER$^{T2}$ recombinase is combined with Cre-activatable reporter transgenes that are driven by ubiquitous promoters. With this approach, stable, inheritable reporter gene expression can be achieved in a distinct cell population labeled by Cre recombination at a predetermined time. Cre reporter transgenes encoding histochemical, fluorescent or bioluminescent reporter proteins have been integrated into the murine Rosa26 (R26) locus, which is accessible to the transcriptional machinery in most if not all cell types[10]. With the currently available R26 Cre reporter mouse lines, however, non-invasive quantitative detection of labeled cells in vivo at the whole-body level is not possible, because detection of the aforementioned reporter proteins relies on either ex vivo methods requiring tissue fixation, invasive methods with a small field of view such as intravital microscopy, or semi-quantitative non-invasive methods such as bioluminescence imaging.

PET is a powerful non-invasive imaging modality in both preclinical and clinical settings. It has a high sensitivity and generates quantitative data, and recent advances in PET-MRI scanner technology enable simultaneous acquisition of functional and morphological information from living mice[11]. Reporter genes for detection of cells by PET cause the accumulation of radiolabeled probes on or in reporter gene-expressing cells[12, 13]. One such PET reporter gene is the herpes simplex virus type 1 thymidine kinase (HSV1-tk). It is used in combination with $^{18}$F- or $^{124}$I-labeled nucleoside analogues, which are phosphorylated by HSV1-tk, but not by mammalian thymidine kinases. In their non-phosphorylated form, PET tracers such as 9-(4[$^{18}$F]-Fluoro-3-[hydroxymethyl]butyl)guanine are "cell-permeable", but after phosphorylation by HSV1-tk they are retained inside the cells. HSV1-tk or an improved variant that enables PET with higher sensitivity, sr39tk[14, 15], have been used for PET imaging of rodents, larger animals and humans[12, 13]. Cre-mediated activation of HSV1-tk expression has been achieved via the delivery of an adenovirus carrying a Cre-activatable HSV1-tk construct to the liver[16] or myocardium[17] of mice expressing Cre in the respective target tissues. However, transgenic mice with a chromosomally integrated Cre-responsive PET reporter gene have not been described to date. In such a mouse line, Cre-expressing cell populations will be labeled for PET imaging through Cre-mediated activation of reporter gene expression at the genomic level. Once reporter gene expression is activated, cells and their progeny are stably labeled, even if the cells proliferate or change their phenotype, which may lead to a loss of Cre expression. This approach would permit non-invasive long-term visualization of any given cell population for which a respective cell type-specific Cre mouse line is available.

To improve cell tracking in mammals, we generated R26 knock-in mice carrying a transgene for Cre-inducible sr39tk expression under control of the ubiquitous cytomegalovirus early enhancer/chicken β-actin/β-globin (CAG) promoter. As these mice express membrane-targeted tandem-dimer tomato red fluorescent protein (mT) before Cre recombination and sr39tk after Cre recombination, we named them "R26-mT/sr39tk" mice. In these mice, a cell population of interest is labeled by Cre-dependent activation of sr39tk expression and then the fate of these cells is followed by non-invasive PET imaging with [$^{18}$F]FHBG. As proof of concept, we demonstrate that the new R26-mT/sr39tk reporter mice enable cell type-specific longitudinal PET imaging of T-cell homing during tissue inflammation and of cardiomyocytes after myocardial infarction (MI). Furthermore, the Cre-responsive PET reporter allele permits non-invasive whole-body characterization of transgenic Cre mouse lines.

## Results

**Generation of R26-mT/sr39tk mice.** We integrated the R26-mT/sr39tk PET reporter construct by homologous recombination into the R26 locus of murine embryonic stem (ES) cells. Before Cre recombination, mT is expressed from the L2 allele, where "L2" stands for "two loxP sites". In Cre-expressing cells, Cre recombinase removes the mT-encoding expression cassette and thereby activates sr39tk expression from the resulting L1 allele ("L1" stands for one loxP site) (Fig. 1 and Supplementary Fig. 1a). Cre-mediated activation of sr39tk expression in ES cells was confirmed at multiple levels: at the DNA level by Southern blot analysis (Supplementary Fig. 1b), at the protein level by western blot analysis (Supplementary Fig. 1c), and at the functional level by testing the sensitivity of the cells to the sr39tk suicide substrate ganciclovir (Supplementary Fig. 1d), as well as by measuring uptake of the PET tracer [$^{18}$F]FHBG into the cells (Supplementary Fig. 1e). ES cells carrying the Cre-responsive L2 allele were used to establish the R26-mT/sr39tk mouse line. In line with previous publications[18, 19], which used the same targeting vector but different reporter genes, we observed strong and ubiquitous mT expression in organs isolated from R26-mT/sr39tk mice (Supplementary Fig. 1f). The general appearance and viability of R26-mT/sr39tk mice (genotype: R26[sr39tk/+], where "+" denotes the wild-type allele) was normal. However, male R26-mT/sr39tk mice showed severely reduced fertility. Male infertility has been described for some other HSV1-tk transgenic mouse lines and is presumably caused by HSV1-tk expression from a cryptic promotor located in the HSV1-tk coding region that is active in postmeiotic germ cells leading to impaired sperm development[20, 21]. Therefore, female R26-mT/sr39tk mice, which showed normal fertility and litter sizes, were used to maintain the R26-mT/sr39tk line and for crossbreeding with transgenic Cre mice.

**Detection of sr39tk-expressing cell populations by PET.** We mated R26-mT/sr39tk mice with Pf4-Cre[22], CD4-Cre[23] or Myh6-Cre[24] mice with Cre[tg/+] genotype (where '+' denotes the wild-type allele) to induce expression of sr39tk in platelets, T lymphocytes or cardiomyocytes, respectively (Fig. 1a and Supplementary Table 1). Animals carrying tissue-specific Cre and R26-mT/sr39tk transgenes are denoted "Cre promoter/sr39tk" mice. We performed in vivo [$^{18}$F]FHBG-PET imaging studies (Fig. 2) with Cre-positive experimental mice that were expected to express sr39tk in the respective target cells (sr39tk+; genotype: Cre[tg/+], R26[sr39tk/+]). To evaluate nonspecific tracer uptake, Cre-negative control animals (sr39tk–; genotype: Cre[+/+],R26[sr39tk/+]) were

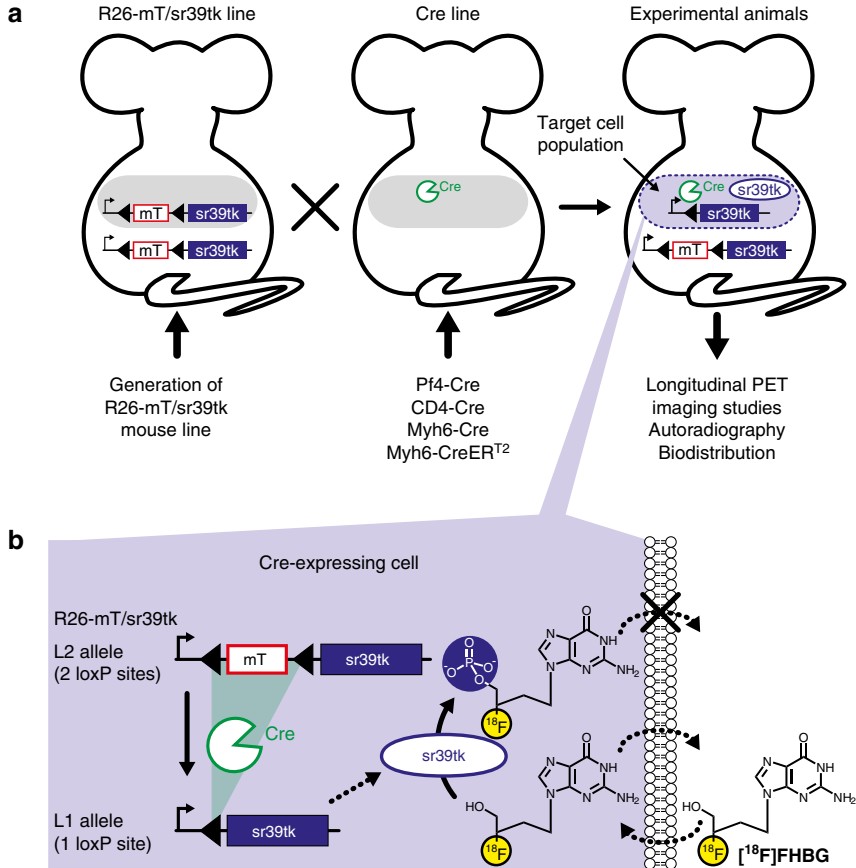

**Fig. 1** Strategy for PET-based cell tracking with the R26-mT/sr39tk mouse line. **a** Generation of experimental animals for PET imaging. R26-mT/sr39tk mice carry the Cre-activatable R26-mT/sr39tk transgene that has been integrated via homologous recombination into the Rosa26 (R26) locus. In R26-mT/sr39tk mice, sr39tk expression is blocked by a loxP-flanked (*triangles*) gene cassette encoding membrane-targeted tandem dimer tomato red fluorescent protein (mT). R26-mT/sr39tk mice are mated with mice that express Cre (or CreER[T2]) under control of a cell type-specific promoter (e.g., Pf4, CD4 or Myh6). In progeny mice, Cre recombination activates sr39tk expression in the respective target cell population (blue shaded oval) as shown in detail in **b**. **b** Strategy for Cre-dependent sr39tk expression and labeling of distinct cell types. The R26-mT/sr39tk L2 allele (carrying two loxP sites) encodes sr39tk preceded by a loxP-flanked (*triangles*) sequence encoding the mT protein. From the L2 allele, mT but not sr39tk is expressed. In cells expressing Cre recombinase, the mT cassette is removed and thereby the L2 allele is converted to the L1 allele (carrying one loxP site), from which sr39tk is expressed under control of the ubiquitous CAG promoter. Sr39tk phosphorylates the tracer molecule [18F]FHBG, which cannot leave the cell once phosphorylated. [18F]FHBG accumulation in sr39tk-expressing cell populations allows their in vivo detection by PET imaging. See also Supplementary Fig. 1

analyzed in parallel. To allocate PET signals to anatomical structures, some animals underwent MRI. In addition to in vivo imaging, [18F]FHBG uptake was assessed *ex vivo* via autoradiography and biodistribution analysis of selected organs (Fig. 3). To validate results obtained with R26-mT/sr39tk PET reporter mice, we tested all Cre transgenes with the well-established R26-lacZ Cre reporter line[10]. LacZ-expressing mice (lacZ+; genotype: Cre[tg/+],R26[lacZ/+]) are denoted "Cre promoter/lacZ" mice. These mice were used to detect Cre-mediated activation of β-galactosidase expression by 5-bromo-4-chloro-3-indolyl-β-D-galactoside (X-Gal) staining of fixed tissues. Cre-negative mice (lacZ−; genotype: Cre[+/+],R26[lacZ/+]) were used as negative controls for X-Gal staining (Supplementary Fig. 2 and 3).

In Pf4/sr39tk mice for visualization of platelets, we observed strong [18F]FHBG accumulation in the spleen and weaker but still significant tracer uptake in lung and skeletal muscle (Fig. 2a, d *left*). Ex vivo analyses confirmed significant tracer accumulation in spleen, lungs and skeletal muscle, and, in addition, revealed sr39tk reporter gene activity in bone marrow and blood (Fig. 3a, d). The activity of the Pf4-Cre transgene in cells of bone marrow, blood and spleen was also demonstrated by detection of β-galactosidase-positive cells in Pf4/lacZ mice (Supplementary Fig. 2a–c). Together, these results indicated that the PET signals

observed in organs of Pf4/sr39tk mice, including signals in lung and skeletal muscle, were indeed derived from megakaryocytes and blood-borne platelets. In agreement with previous reports, nonspecific accumulation of background radioactivity was observed in the gastrointestinal tract[25] as well as in bone[26] of sr39tk+and sr39tk− animals (Fig. 2a–c).

CD4/sr39tk mice for T-cell tracking studies showed specific [18F]FHBG uptake in the spleen, lymph nodes and thymus, and, unexpectedly, in the heart and lung (Figs. 2b, d *middle* and 3b, e). The PET signal observed in the spleen of CD4/sr39tk mice was significantly stronger than in sr39tk− control mice, but weaker than in Pf4/sr39tk mice (Figs 2d and 3d, e). In line with these results, CD4/lacZ mice demonstrated β-galactosidase activity in lymphatic organs (Supplementary Fig. 2d–f) as well as in heart and lung (Supplementary Fig. 2g, h). Closer inspection of X-Gal-stained tissue sections and immunostaining of marker proteins indicated previously unknown ectopic activity of the CD4-Cre transgene in smooth muscle cells of medium-caliber vessels of the heart (Supplementary Fig. 3a, b) as well as in bronchial epithelial cells of the lung and alveolar macrophages (Supplementary Fig. 3c–f). Ectopic activity of the CD4-Cre line in some vascular smooth muscle cells could also explain the weak but significant tracer uptake that was detected ex vivo in skeletal muscle (Fig. 3e).

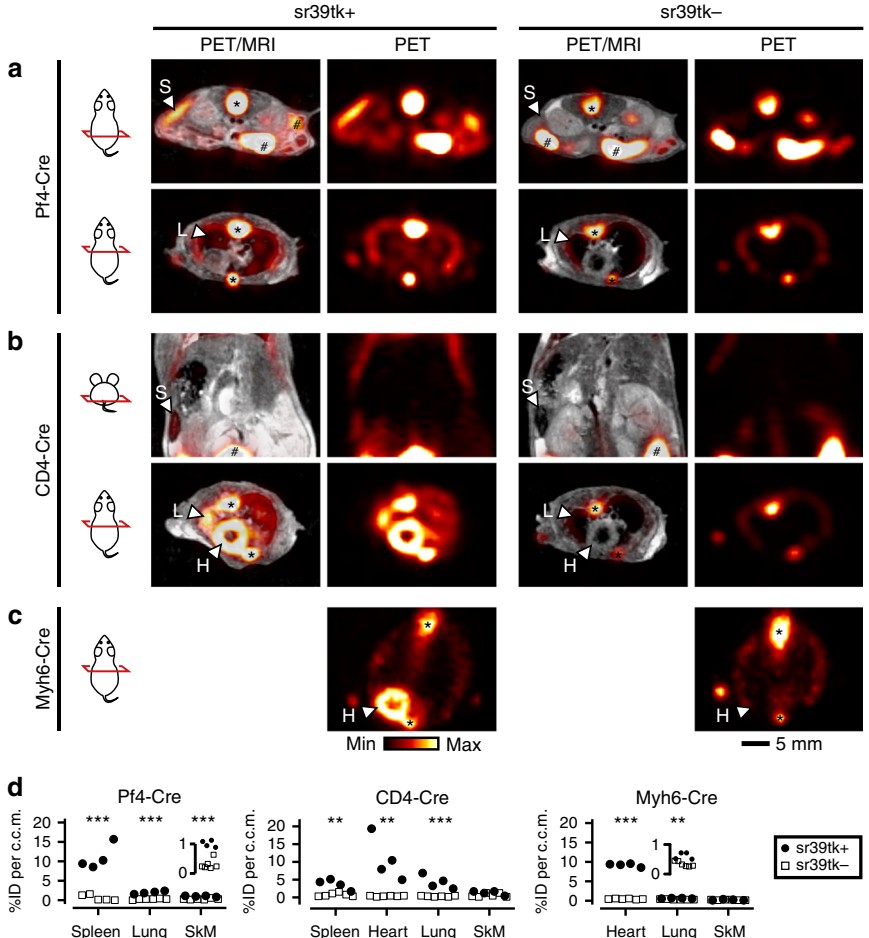

**Fig. 2** In vivo analysis of [18F]FHBG uptake in mice with different sr39tk-expression profiles. Panels show representative [18F]FHBG-PET images of sr39tk-expressing mice (sr39tk+; genotype: Cre[tg/+],R26[sr39tk/+]) and Cre-negative control animals (sr39tk−; genotype: Cre[+/+],R26[sr39tk/+]); '+' denotes the wild-type allele. To confirm uptake into distinct anatomic locations, MR images were recorded and overlaid with PET images (PET/MRI). Nonspecific signals that were observed in both sr39tk+ and sr39tk− animals originate from the bone (*) and gastrointestinal tract (#). Specific tracer uptake was detected in **a** Pf4/sr39tk mice (Pf4-Cre) in the spleen (S), lung (L) and skeletal muscle (see also **d** *left*); **b** CD4/sr39tk mice (CD4-Cre) in the spleen (S), heart (H) and lung (L) (see also **d** *middle*); **c** Myh6/sr39tk mice (Myh6-Cre) in the heart (H) and lung (see also **d** *right*). PET images of sr39tk+ mice and sr39tk− control animals were normalized to injected tracer dose. Pictograms on the left show orientation and relative position of the image plane (*red*). **d** [18F]FHBG uptake was quantified from manually drawn regions of interest (ROIs) in selected organs of four Pf4/sr39tk and six sr39tk− mice (*left*), four CD4/sr39tk and six sr39tk− mice (*middle*), and four Myh6/sr39tk and six sr39tk− mice (*right*) (SkM, skeletal muscle). Data points represent individual animals. Tracer uptake was normalized to injected tracer dose (ID) and ROI volume (in c.c.m.). Insets for skeletal muscle (Pf4-Cre) and lung (Myh6-Cre) show [18F]FHBG uptake between 0 and 1 %ID per c.c.m. One-way ANOVA was performed to compare tracer uptake into organs of sr39tk+ and sr39tk− animals **p < 0.01 and ***p < 0.001, respectively). See also accompanying ex vivo analysis of [18F]FHBG uptake in Fig. 3, further analysis of Cre recombinase activity in Supplementary Figs 2 and 3, and characterization of the effects of sr39tk expression on cell counts in Supplementary Fig. 4

In Myh6/sr39tk mice for detection of cardiomyocytes, in vivo PET imaging (Fig. 2c, d *right*) as well as ex vivo analysis of tissues (Fig. 3c, f) revealed strong [18F]FHBG uptake in the heart and, to a much smaller extent, in the lungs. These results were confirmed in Myh6/lacZ mice, which showed β-galactosidase activity in cardiomyocytes (Supplementary Fig. 2i) as well as in pulmonary vascular smooth muscle cells (Supplementary Figs. 2j and 3g, h).

We also explored whether sr39tk expression exerted effects on the respective target cells. The general appearance and viability of Cre/sr39tk mice (genotype: Cre[tg/+],R26[sr39tk/+]) analyzed in this study was normal. In Pf4/sr39tk mice, platelet counts in peripheral blood were not significantly different from control mice (Supplementary Fig. 4a). In naive CD4/sr39tk mice (genotype: Cre[tg/+],R26[sr39tk/+]), we determined T-cell numbers in the lymph nodes, spleen and thymus by flow cytometry. Compared with wild-type mice (genotype: Cre[+/+], R26[+/+]), CD4/sr39tk mice had a smaller number of CD3+

lymphocytes, which was primarily due to a smaller fraction of CD4+ T cells (Supplementary Fig. 4b–d). It has been reported that CD4-Cre mice have reduced T-cell numbers, particularly CD4+ T cells in the spleen[27]. However, the CD4-Cre mice (genotype: Cre[tg/+],R26[+/+]) used in our studies showed T-cell numbers similar to wild-type mice, except for a slightly lower number of CD8+ T cells in the thymus that was statistically significant (Supplementary Fig. 4b–d). Thus, the reduced number of T cells we observed in CD4/sr39tk mice was apparently mainly caused by expression of sr39tk and not by expression of Cre recombinase. The spleen weight of CD4/sr39tk mice was not altered compared with control mice (Supplementary Fig. 4e).

Taken together, these results indicated that the R26-mT/sr39tk mouse line is useful to label a broad spectrum of cell types via crossbreeding to existing tissue-specific Cre lines. Importantly, the expression of the sr39tk reporter gene after Cre-mediated activation was strong enough to enable non-invasive detection of the cell

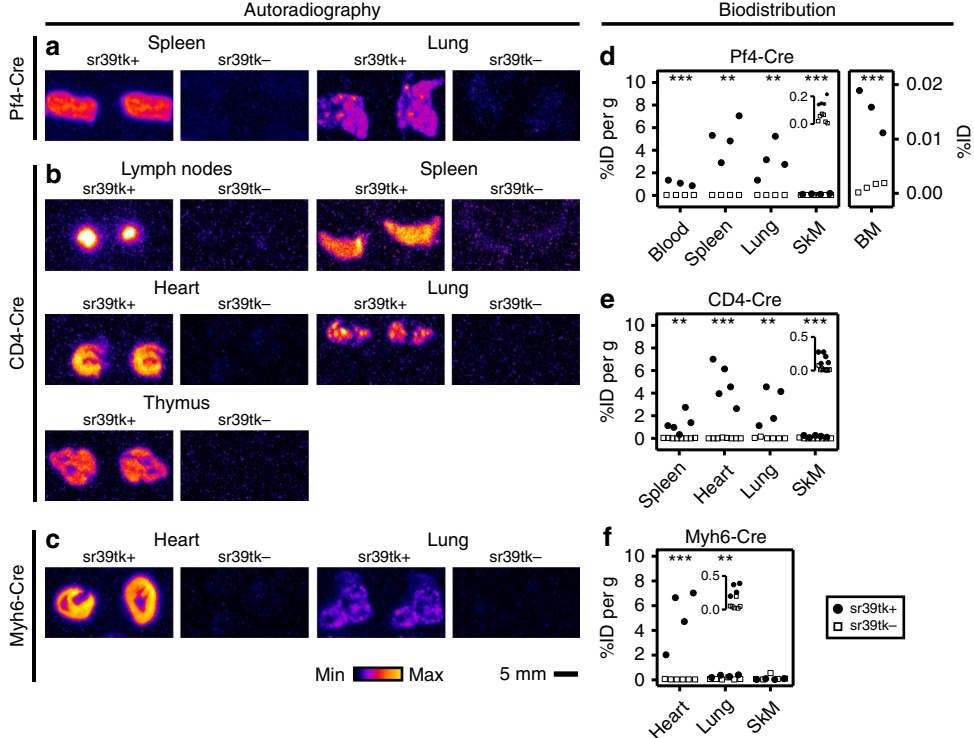

**Fig. 3** Ex vivo analysis of [18F]FHBG uptake in mice with different sr39tk expression profiles. **a–c** Representative [18F]FHBG autoradiographs from various organs of (**a**) Pf4/sr39tk (Pf4-Cre), (**b**) CD4/sr39tk (CD4-Cre) and (**c**) Myh6/sr39tk (Myh6-Cre) mice. sr39tk-expressing mice (sr39tk+; genotype: Cre[tg/+],R26[sr39tk/+]) were compared with Cre-negative control mice (sr39tk−; genotype: Cre[+/+],R26[sr39tk/+]); '+' denotes the wild-type allele. Twenty-micrometer cryosections were used except for lymph nodes (**b**), which were not cut before autoradiography. Autoradiographs of the same organs from sr39tk+ and sr39tk− mice were derived from the same phosphor screen, but they were not normalized between organs or Cre lines. Organs were not cleared from blood before analysis. Similar results were obtained with organs from ≥3 animals of each genotype. **d–f** Biodistribution analysis of [18F]FHBG uptake into organs isolated from (**d**) Pf4/sr39tk (Pf4-Cre), (**e**) CD4/sr39tk (CD4-Cre) and (**f**) Myh6/sr39tk (Myh6-Cre) mice with (sr39tk+, *black symbols*) or without (sr39tk−, *open symbols*) expression of sr39tk. Uptake was normalized to injected tracer dose (ID) and tissue weight, except for bone marrow (BM, **d**), which was flushed from one tibia and femur per mouse with 1 mL PBS. Data points represent individual animals. Inset for skeletal muscle in **d** shows uptake between 0 and 0.2 %ID per g tissue and insets for skeletal muscle in **e** and lung in **f** show [18F]FHBG uptake between 0 and 0.5 %ID per g tissue. One-way ANOVA was used to compare [18F]FHBG uptake into organs of sr39tk+ and sr39tk− animals (**p < 0.01 and ***p < 0.001, respectively). Organs were not cleared from blood before analysis. BM, bone marrow; SkM, skeletal muscle

population of interest and of previously unknown Cre activity in other cell types, via non-invasive PET imaging in vivo. Next, we evaluated the suitability of the R26-mT/sr39tk reporter mouse line for longitudinal cell tracking in two clinically relevant disease models.

**T-cell homing during inflammation**. To monitor T-cell homing during inflammation, we used a mouse model of allergic contact dermatitis, one of the most frequent inflammatory skin diseases. It is characterized by a T-cell-mediated hypersensitivity reaction of the skin and also studied as a paradigm for autoimmune disorders, antiviral and antitumor immunity[28]. A cutaneous delayed-type hypersensitivity reaction (DTHR) was induced in CD4/sr39tk (sr39tk+) mice and, as a control for nonspecific tracer uptake, in Cre-negative mice (sr39tk−). In our animal model[29], mice were first sensitized at the abdomen by application of 2,4,6-trinitrochlorobenzene (TNCB) at day 0. Then, we elicited a cutaneous DTHR on the left ear by repetitive application of TNCB on day 7, 10 and 12, and used [18F]FHBG-PET to follow homing of sr39tk-expressing T lymphocytes to the left ear. PET imaging was performed on day 6 (1 day before the first TNCB challenge), on day 13 (1 day after the last TNCB challenge) and on day 20 (Fig. 4a). The degree of inflammation was determined by measuring ear thickness before and after TNCB ear challenges (Supplementary Fig. 5a). On day 13, 12 h after the last TNCB ear challenge, PET imaging of the animals revealed strong

accumulation of [18F]FHBG in the challenged left ear, but not in the non-challenged right ear of sr39tk+ mice or in ears of sr39tk− control animals (Fig. 4b). Compared with baseline on day 6, [18F]FHBG uptake into the TNCB-challenged left ears of sr39tk+ mice was ≈6-fold higher in each animal on day 13 and then returned to a lower level on day 20 (Fig. 4b right), in line with reduced inflammation and ear thickness 8 days after the third TNCB- challenge (Supplementary Fig. 5a). In contrast, [18F]FHBG uptake into the non-challenged right ears of sr39tk+ mice or into the ears of sr39tk− animals remained at a low level over the entire time course (Fig. 4b right). PET imaging of the spleen, lung and liver did not show significant changes in tracer uptake over time in these organs, indicating the absence of a strong systemic immune response in the experimental animals (Supplementary Fig. 5b–d). Autoradiography of the ears on day 20 confirmed strong [18F]FHBG uptake into the TNCB-challenged left ears of sr39tk+ mice (Fig. 4c). Autoradiography detected elevated tracer uptake also in the non-challenged right ears of sr39tk+ mice, in particular in one of the three sr39tk+ animals analyzed (Fig. 4c). This was likely due to scratching and transfer of TNCB from the left to the right ear, thereby, inducing inflammation also in the "non-challenged" right ear. The data obtained in vivo by PET imaging and ex vivo by autoradiography showed a good correlation (Supplementary Fig. 5e). We can exclude that the increased [18F]FHBG uptake in TNCB-

                    5

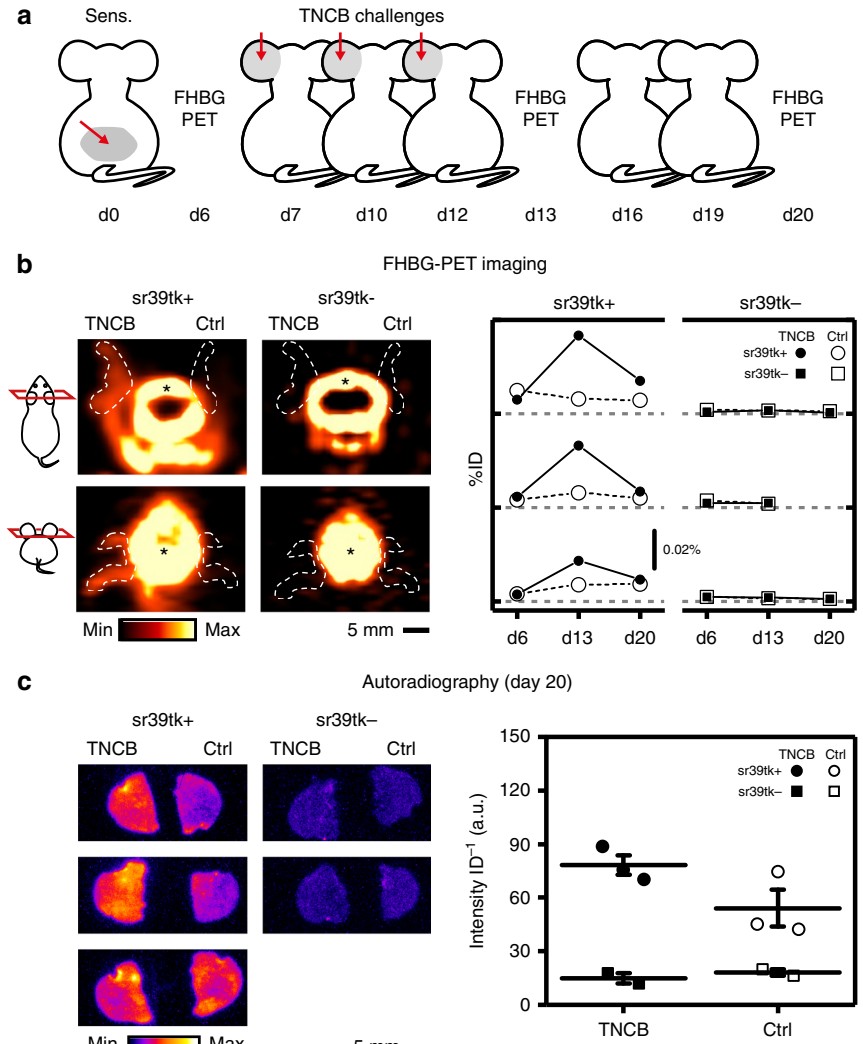

**Fig. 4** Longitudinal tracking of T-cell homing in CD4/sr39tk mice during cutaneous DTHR. **a** At day 0 (d0), four CD4/sr39tk mice (sr39tk+; genotype: CD4-Cre[tg/+],R26[sr39tk/+]) and three litter-matched Cre-negative control animals (sr39tk−; genotype: CD4-Cre[+/+],R26[sr39tk/+]) were sensitized by cutaneous application of 5% TNCB to the abdomen (*red arrow* at day d0). Cutaneous DTHR on the left ear was elicited by three repetitive challenges with 1% TNCB on day 7, 10, 12 (*red arrows*). Animals were analyzed by [18F]FHBG-PET on day 6, 13 and 20, and killed on day 20 for ex vivo analysis of [18F]FHBG uptake. One sr39tk+ animal developed an infection of the TNCB-challenged ear and was excluded from analysis. One sr39tk− animal died on day 20. **b** *Left*, representative [18F]FHBG-PET images of the head region of a sr39tk+ and a sr39tk− mouse recorded on day 13. Ear boundaries are outlined by dashed lines; left ears had been challenged with TNCB, whereas right ears had not been challenged (Ctrl). Images were normalized to injected tracer dose. Pictograms on the left show orientation and relative position of the image plane (*red*). Nonspecific tracer accumulation was detected in bone (*). *Right*, longitudinal PET imaging showing the time course of [18F]FHBG uptake into TNCB-challenged ears (TNCB, black symbols) and non-challenged ears (Ctrl, white symbols) of individual sr39tk+ (left) and sr39tk− (right) mice. Values were normalized to injected tracer dose (ID); *grey dashed lines* represent 0% ID for each animal. **c** *Left*, [18F]FHBG autoradiography of the ears from three sr39tk+ and two sr39tk− mice at the end of the study on day 20; left ears had been challenged with TNCB; right ears had not been challenged (Ctrl). *Right*, quantification of [18F]FHBG uptake by autoradiography of the TNCB-challenged ears (TNCB, *black symbols*) and non-challenged ears (Ctrl, *white symbols*). Intensity (arbitrary units, a.u.) was background-corrected and normalized to injected tracer dose (ID); data points indicate individual animals; bars represent mean ± s.e.m. See also Supplementary Fig. 5

challenged ears of sr39tk+ mice was the consequence of inflammation-induced increases in perfusion or vascular permeability (Supplementary Fig. 5a) as we observed no [18F]FHBG uptake in repetitively TNCB-challenged and, thus, strongly inflamed ears of Cre−, sr39tk− control animals (Fig. 4b, c). In sum, our data obtained with CD4/sr39tk mice in the DTHR inflammation model demonstrated that the novel PET reporter transgene allows non-invasive in vivo tracking of endogenous T cells longitudinally over several weeks in mice.

**Cardiomyocyte viability after MI**. As a second clinically relevant disease model, we performed longitudinal PET imaging of cardiomyocytes after MI in Myh6i/sr39tk mice. Myh6i mice express the tamoxifen-inducible CreER^T2 recombinase under control of the cardiomyocyte-specific Myh6 promoter[30]. To activate sr39tk expression specifically in cardiomyocytes of adult mice, 17-week-old mice received 1 mg tamoxifen per day for 5 days by intraperitoneal injection. Four weeks later, MI was induced by 60 min temporary ligation (ischemia/reperfusion, I/R group) or permanent ligation (Lig group) of the left anterior descending coronary artery (LAD). Control animals underwent surgery without LAD ligation (Sham group). The study protocol is depicted in Fig. 5a. In week 1, 2 and 3 after MI, we performed [18F]FHBG-PET imaging to monitor viable cardiomyocytes. In

the same animals, glucose uptake into the heart was detected by 2-[18F]fluoro-2-deoxy-D-glucose ([18F]FDG)-PET imaging in week 1 and 2 after MI. To suppress [18F]FDG uptake into cardiomyocytes and allow imaging of post-infarct inflammation, we used ketamine and xylazine for anesthesia. With this protocol it is possible to preferentially detect non-cardiomyocytes that take up [18F]FDG, such as immune cells (neutrophils, macrophages, etc.) known to accumulate in the post-ischemic myocardium[31, 32]. To analyze regional differences in tracer uptake in sham-operated and infarcted hearts, we created polar maps (PMs) of the left ventricle. Representative PET images and corresponding PMs are shown in Fig. 5b (for an extended presentation of PET data, see Supplementary Fig. 6). As expected, sham-operated Myh6i/sr39tk mice showed strong and uniform [18F]FHBG uptake into the myocardium, whereas [18F]FDG uptake was low (Fig. 5b upper).

Infarcted hearts of the I/R group (Fig. 5b middle) and Lig group (Fig. 5b lower) could be well imaged by detection of viable cardiomyocytes with [18F]FHBG-PET, with the respective infarcted area spared. [18F]FHBG-PET signals appeared stable over the whole time course of ≈3 weeks after MI. Compared with remote myocardium, [18F]FDG uptake into the infarct zone appeared to be increased after MI, presumably due to uptake by infiltrating immune cells. In contrast to the results with [18F]FHBG-PET the [18F]FDG-PET signals in infarcted hearts appeared to be more variable over time, and in mice of the I/R group [18F]FDG uptake was clearly present in the infarcted regions spared by [18F]FHBG (Fig. 5b).

Quantification of tracer uptake into individual segments of the left ventricle confirmed differences between [18F]FHBG-PET and [18F]FDG-PET signals (for an illustration of the segmentation model, see Supplementary Fig. 7a). The infarcted area was defined as segments with a reduction of [18F]FHBG uptake to <50% of maximal uptake in the same heart. The number of infarcted segments was similar in the I/R group and the Lig group (Supplementary Fig. 7b). Uptake of [18F]FHBG into the whole left ventricle (infarcted and non-infarcted segments) as well as into the infarcted area alone was stable over 18 days after MI (Supplementary Fig. 7c left, d left). In contrast, uptake of [18F]FDG into the whole left ventricle as well as infarcted area was higher at 5 days than at 14 days after MI, particularly in the I/R group (Supplementary Fig. 7c right, d right). In both the I/R and Lig group, [18F]FHBG uptake into the infarcted area was consistently reduced by 50–60% compared to the non-infarcted area of the left ventricle (Supplementary Fig. 7e left). [18F]FDG uptake into infarcted areas in the Lig group was reduced by ≈25%, whereas infarcted areas from mice in the I/R group showed similar or even slightly higher [18F]FDG uptake than non-infarcted myocardium (Supplementary Fig. 7e right). Some Myh6i/sr39tk animals in this study also carried the R26-lacZ Cre reporter transgene. This enabled us to correlate in vivo

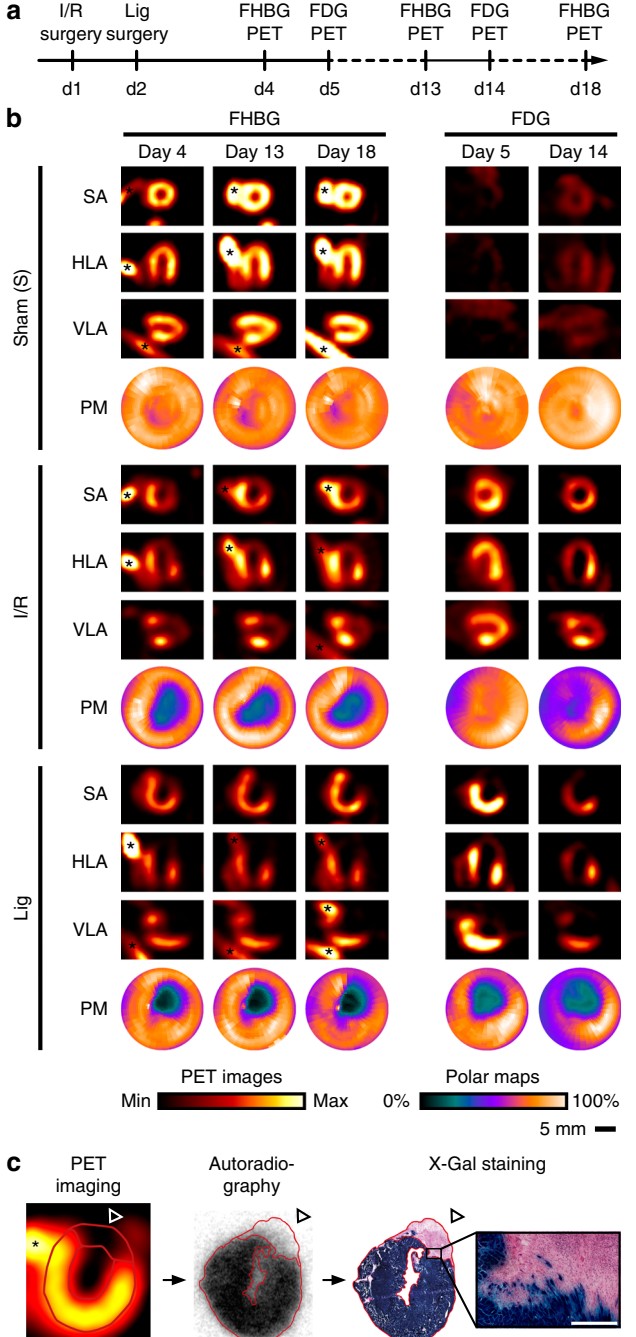

**Fig. 5** Longitudinal [18F]FHBG and [18F]FDG PET study in Myh6i/sr39tk mice with myocardial infarction (MI). **a** Expression of sr39tk was induced in cardiomyocytes of 11 Myh6i/sr39tk mice (genotype: Myh6i-Cre[tg/+], R26[sr39tk/+]) by intraperitoneal injection of 1 mg tamoxifen for 5 consecutive days starting 4 weeks before surgery. MI was caused on day 1 (d1) by 60-min temporary ligation (ischemia/reperfusion, I/R) or on day 2 by permanent ligation (Lig) of the left anterior descending coronary artery in three to five animals per group. Two animals underwent surgery without ligation (Sham). All mice underwent [18F]FHBG-PET on day 4, 13 and 18, and [18F]FDG-PET on day 5 and 14. **b** Representative PET images of the heart (HLA, horizontal long axis; SA, short axis; VLA, vertical long axis) and corresponding polar maps (PMs) of the left ventricle of individual mice over the time course of the study (for an extended presentation of PET images, see Supplementary Fig. 6). PMs provide an overview of myocardial tracer uptake and were segmented for further regional quantification (see Supplementary Fig. 7). Nonspecific signals were derived from bone (*). PET images were normalized to injected dose of the respective tracer, PMs were normalized to maximal tracer uptake for each tracer, animal and image acquisition. **c** Some Myh6i/sr39tk mice of this study carried also the R26-lacZ Cre reporter transgene. A representative example of a mouse with permanent ligation is shown. After in vivo [18F]FHBG-PET imaging (left, nonspecific signals from bone are indicated by *) and autoradiography of a 20-μm short-axial cryosection of the heart (middle) on day 18, β-galactosidase activity was detected in the same section via X-Gal staining (right, counterstain with hematoxylin and eosin). Red outlines demonstrate the correlation between loss of [18F]FHBG uptake and loss of β-galactosidase activity in the infarcted region (open arrowheads). High-magnification image shows the border region between healthy and infarcted tissue at single-cell resolution (enlarged from overview picture; scale bar: 250 μm). See also Supplementary Figs 6 and 7

[18F]FHBG-PET imaging and ex vivo [18F]FHBG-autoradiography with β-galactosidase activity in identical heart sections. We observed that loss of [18F]FHBG uptake correlated well with loss of β-galactosidase activity (Fig. 5c). In sum, these experiments showed the feasibility of longitudinal [18F]FHBG-PET imaging using our cardiomyocyte-specific sr39tk reporter mice to monitor cardiomyocyte viability after MI and to characterize infarcts in detail, particularly when [18F]FHBG- and [18F]FDG-PET are combined to image cardiomyocytes and immune cells, respectively.

## Discussion

In this study, we have established a modular Cre/lox-based system for non-invasive labeling, tracking and quantification of distinct cell populations in mice. This system relies on stable expression of the sr39tk PET reporter in the cells of interest as well as their progeny. In addition to enabling in vivo cell tracking at the whole-body level, our approach has several advantages over cell-tracking methods that are based on direct cell labeling or expression of a reporter transgene driven by a cell-type-specific promoter. As the sr39tk transgene is integrated into the genome, it permits stable cell labeling that is passed on to all progeny cells. Initially, expression of sr39tk is silenced, but it can be activated in the cell type of interest by crossbreeding with an appropriate cell-type-specific Cre mouse line. Once activated, sr39tk expression is driven by the ubiquitous CAG promoter, rendering it independent of subsequent changes of cell differentiation and potential silencing of tissue-specific promoters. Furthermore, no ex vivo labeling procedures are required, i.e., the cell population of interest is labeled in situ in its physiological environment.

As a proof of concept, we labeled and visualized various cell types by PET imaging and we validated our new mouse model for longitudinal non-invasive cell tracking in two clinically relevant disease models. In Pf4/sr39tk, CD4/sr39tk or Myh6/sr39tk mice, we were able to detect platelets, T cells or cardiomyocytes, respectively, by [18F]FHBG-PET imaging in vivo. In general, expression of sr39tk did not cause gross abnormalities in the number and/or function of the target cells. We noted that the number of CD4+ T cells was reduced in CD4/sr39tk mice as compared with control mice, but the TNCB-specific response of sr39tk-expressing T cells in the context of our DTHR model was apparently not compromised, as the extent of ear swelling in CD4/sr39tk mice was similar to control mice not expressing sr39tk. The reason for the lower T-cell number in CD4/sr39tk mice is not clear. It is unlikely that it was caused by the potential immunogenicity of the sr39tk protein[33], because CD4/sr39tk mice express the protein permanently and should, therefore, not mount an immune response against it.

Platelets, T cells and cardiomyocytes could be visualized in vivo in anesthetized mice and specific [18F]FHBG uptake in these cells was confirmed by ex vivo analyses. In few cases, in vivo detection of the respective target cells was compromised by nonspecific accumulation of radioactivity and/or unexpected Cre activity in some tissues. For instance, it was difficult to detect T cells in distinct lymph nodes and the thymus of CD4/sr39tk mice in vivo. Visualization of lymph nodes was impeded by background radioactivity in the bone and gastrointestinal tract, whereas the thymus was masked by [18F]FHBG accumulation in the heart and lungs. The latter was due to previously unknown activity of the CD4-Cre transgene in non-lymphocyte cells of heart and lung. The problem of ectopic Cre activity can be solved by using Cre lines with alternative promoters to drive Cre expression. The current limitation in quantifying T cells in lymph nodes could be overcome by increasing the purity of the [18F]FHBG preparation, which was 82–92% in the present study, thereby reducing the

amount of free [18F]fluoride, which accumulates in the bone. Background radioactivity in the gastrointestinal tract could be reduced by increasing [18F]FHBG elimination via the intestine or by using alternative sr39tk substrates with faster clearance such as [18F]FEAU[34, 35]. In addition, the use of thymidine kinase substrates carrying [124I] as longer-lived PET isotope has been described[35, 36]. With [124I]-labeled radiotracers, PET imaging can be performed much later after tracer injection than with short-lived [18F]-labeled tracers, thereby allowing more 'non-trapped' tracer to be eliminated before image acquisition. Another approach to reduce background signals could be the use of alternative PET reporter genes such as the human or rat sodium iodide symporter. However, the respective tracer, iodine, may show the problem of insufficient intracellular retention due to iodine efflux[37].

Detailed knowledge of the specificity of Cre recombination is not only important for cell tracking experiments as described in the present study, but also when Cre mice are used to generate tissue-specific gene knockouts. Published characterization data for Cre strains are often limited to a few specific tissues that were studied in the context of a focused research question. However, it is increasingly recognized that any given transgenic Cre line may display recombination activity beyond the intended tissue or cell type. Therefore, careful characterization of Cre mouse lines is warranted[38]. The R26-mT/sr39tk reporter mouse provides the possibility for recombination profiling of transgenic Cre mice on a whole-body scale using PET imaging. This novel approach complements existing strategies for comprehensive Cre strain characterization that are based on post-mortem analyses of conventional LacZ Cre reporter mice (http://www.creportal.org)[38]. Indeed, using our sr39tk PET reporter mice for whole-body imaging combined with the LacZ Cre reporter for subsequent post-mortem analyses of tissue sections, we have identified several previously unreported activities of commonly used Cre strains. Ectopic activity was detected in CD4-Cre mice in smooth muscle cells of myocardial blood vessels as well as in alveolar macrophages and bronchial epithelial cells of the lung, and in Myh6-Cre mice in the lung vasculature.

Among the most exciting applications of our novel sr39tk PET reporter mouse are longitudinal cell tracking studies in the context of health, disease and therapy. For example, tracking of T lymphocytes on the whole-body level can improve our understanding of immune responses to infections and malignancies or during immunotherapy. Diverse procedures have been developed to label T cells in vitro for subsequent PET imaging in mice and humans[39–41]. However, these experiments are often limited by label dilution and artificial signal loss caused by the proliferation of T cells in the course of an immune response[42]. In CD4/sr39tk mice we were able, without the need for in vitro labeling, to follow the homing of endogenous CD4+ T lymphocytes to sites of TNCB-specific tissue inflammation over several weeks.

In another clinically relevant disease model, we used Myh6i/sr39tk mice to monitor cardiomyocyte viability after MI. In both preclinical and clinical studies, [18F]FDG is often used for this purpose. However, [18F]FDG reports cellular glucose uptake and metabolism, which are subject to complex regulation and alterations under pathophysiological conditions such as ischemia[43]. Moreover, uptake of [18F]FDG is not cell-type specific. In addition to cardiomyocytes, other cells in the myocardium may accumulate [18F]FDG, in particular immune cells in the context of post-infarct inflammation[31, 32]. For these reasons, detection of sr39tk-labeled cardiomyocytes with [18F]FHBG appears to be a more robust approach than [18F]FDG imaging, to specifically assess cardiomyocyte viability in animal models of cardiac disorders. Thus, we sought to use [18F]FHBG-PET for specific detection of cardiomyocytes and [18F]FDG under conditions that

favor its uptake into immune cells over cardiomyocytes for PET imaging of post-infarct inflammation. Sr39tk or related PET reporters have previously been expressed in rodent cardiomyocytes using adenoviral approaches, but expression was limited to the adenovirus-infected region and began to vanish some days after virus administration[17, 44, 45]. In contrast, in Myh6i/sr39tk mice we observed uniform and stable sr39tk expression in the entire myocardium, which allowed longitudinal tracking of cardiomyocyte viability in sub-regions of the heart after MI. The spatial profile and relative amount of [18F]FHBG uptake after MI induction remained constant over the whole 3-week period of our study. The combination of PET imaging with [18F]FHBG and [18F]FDG enabled us to characterize infarcts in detail by visualization of viable cardiomyocytes and post-infarct inflammation, respectively. The fact that permanent LAD ligation results in almost complete loss of perfusion of the infarcted zone might complicate the interpretation of tracer uptake in this region after permanent LAD ligation. However, after I/R injury, perfusion is maintained and thus it appears that the stable reduction of [18F]FHBG uptake in the infarcted region was indeed due to cardiomyocyte death rather than loss of perfusion. In contrast, [18F]FDG uptake after I/R injury was initially increased in the infarcted region and was more variable over time, presumably due to inflammatory processes within the infarcted region and [18F]FDG uptake into immune cells.

Taken together, this study demonstrates the suitability of our novel sr39tk PET reporter mouse for non-invasive visualization, quantification and tracking of specific cell populations in mice. The binary nature of the system permits labeling of a broad spectrum of cell types by crossbreeding the sr39tk reporter line with different Cre strains, several hundreds of which are available for targeting basically any cell type of interest (http://www.creportal.org). It is also possible to combine the sr39tk PET reporter allele with other Cre-switchable reporter transgenes. This enables, for instance, combined in vivo imaging of cells by PET and other modalities (e.g., fluorescence, bioluminescence) or validation of PET data by post-mortem histochemical cell detection by X-Gal staining of β-galactosidase activity. Moreover, combination of the sr39tk PET reporter with Cre-sensitive conditional knockout alleles allows in vivo-tracking of cell populations that lack a protein of interest. Previous methods for cell tracking with PET were based on direct in vitro-labeling of cells or transgenic expression of a PET reporter protein under the control of a tissue-specific promoter. Our Cre/lox-based PET reporter approach has one major advantage over these methods, in that it permits in vivo labeling of essentially any cell type of interest followed by stable/inheritable expression of the PET reporter protein irrespective of cell phenotype and potential silencing of a tissue-specific promoter. The latter property might be particularly useful to keep track of initially labeled cells after dedifferentiation or transdifferentiation, for instance, during vascular remodeling[46] or cell therapy[3]. If tamoxifen-inducible Cre recombinases such as CreER[T2] are used for cell labeling, it should be considered that only cells will be labeled that expressed the recombinase during the tamoxifen pulse and that these cells might later be "diluted" by non-labeled cells originating from a pool of CreER[T2]-negative progenitor cells. For instance, if in our MI model newly formed cardiomyocytes were derived from sr39tk– progenitor cells, then these cardiomyocytes would not take up [18F]FHBG, and cell viability would be underestimated by [18F]FHBG-PET. In general, labeling of rapidly proliferating cell populations with tamoxifen-inducible CreER[T2] may be mosaic and, therefore, non-recombined cells may dilute the reporter signal when they proliferate. This problem can be addressed by using a non-inducible Cre line, for instance, the Myh6-Cre line for efficient labeling of cardiomyocytes.

We foresee many applications of our sr39tk PET reporter mice in preclinical research. In combination with MRI or other PET tracers (as shown in the present study), this mouse line will improve our understanding of mammalian (patho-)biology associated with migration, accumulation, death, or survival of distinct cell populations. These processes are of fundamental importance in clinical conditions such as inflammation, diabetes, atherosclerosis, thrombosis, MI, stroke and tumorigenesis. Our method will also be useful to elucidate endogenous mechanisms of tissue degeneration and regeneration as well as effects of therapeutic interventions. Cells derived from sr39tk PET reporter ES cells or mice can aid the development of effective cell-based therapies, which requires monitoring of the location, distribution and long-term viability of the transplanted cells in a non-invasive manner[3, 4]. In this context, sr39tk can be used not only as a PET reporter, but also as a suicide gene enabling the elimination of therapeutic cells by ganciclovir treatment, if they are causing severe adverse effects[47]. We anticipate that the Cre-switchable sr39tk PET reporter mice will be broadly applicable to address complex biological questions in vivo and, compared to invasive cell-tracking methods, enable preclinical research with fewer animals and complementary informative value.

## Methods

**Experimental animals.** All animal experiments were performed at the University of Tübingen and approved by the local authority (Regierungspräsidium Tübingen, PC 1/08 and IB 1/13). Mice were housed in a barrier-free or individually-ventilated-cage mouse facility at 19–22 °C and 40–60% humidity in a 12/12 h light/dark cycle with free access to standard rodent chow and tap water. R26-mT/sr39tk PET reporter mice (B6;129-Gt(ROSA)26Sor^tm2(ACTB-tdTomato,-sr39tkFeil)) were generated in this study and had a mixed 129 Sv/C57BL6N genetic background. R26-lacZ Cre reporter mice (B6.129-Gt(ROSA)26Sor^tm1Sor)[10] were on a C57BL/6N background. Cre mice were obtained from the following sources: CD4-Cre (Tg(Cd4-cre)^1Cwi)[23] from Ari Waisman (University of Mainz, Mainz, Germany); Pf4-Cre (Tg(Cxcl4-cre)^Q3Rsko/J)[22] and Myh6-Cre (Tg(Myh6-cre)^2182Mds)[24] from The Jackson Laboratories (Bar Harbor, ME, USA); Myh6-CreER^T2 (Tg(Myh6-icre/ERT2)^Wet)[30] from Andreas Friebe (Universität Würzburg, Würzburg, Germany). All Cre lines were maintained on a C57BL/6N background.

The generate experimental animals, male Cre[tg/+],R26[+/+] mice were mated with female Cre[+/+],R26[sr39tk/+] mice; "+" denotes the wild-type allele. To correlate [18F]FHBG accumulation with expression of β-galactosidase encoded by the R26-lacZ Cre reporter[10], male Cre[tg/+],R26[lacZ/+] mice were mated with female Cre[+/+],R26[sr39tk/+] mice. Experimental animals were males and females with an age between 9 and 42 weeks (for further details, see Supplementary Table 1). The sr39tk-expressing mice (designated Cre/sr39tk or sr39tk+) had the genotype Cre[tg/+],R26[sr39tk/+]; animals that expressed sr39tk and β-galactosidase (designated Cre/sr39tk+ lacZ) had the genotype Cre[tg/+], R26[sr39tk/lacZ]; Cre-negative control animals that did not express sr39tk (designated sr39tk–) or lacZ (designated lacZ–) had the genotype Cre[+/+], R26[sr39tk/+] or Cre[+/+],R26[sr39tk/lacZ]. In addition, Cre[tg/+] mice were mated with R26[lacZ/+] mice to generate β-galactosidase-expressing animals (designated Cre/lacZ; genotype: Cre[tg/+],R26[lacZ/+]) and respective Cre-negative control animals (genotype: Cre[+/+],R26[lacZ/+]).

**Generation of R26-mT/sr39tk mice.** The gene targeting vector pRosa26-mT/sr39tk was used to integrate the CAG promoter, a loxP-flanked expression cassette for membrane-targeted tandem-dimer red fluorescent protein (mT), and the sr39tk encoding sequence followed by a FRT-flanked neomycin-resistance cassette into the murine R26 locus (Supplementary Fig. 1a). To generate pRosa26-mT/sr39tk, the coding sequence of membrane-targeted green fluorescent protein (mG) in pRosa26-mT/mG[18] (Addgene plasmid 17787) was replaced by the sr39tk-encoding sequence[15] isolated from pCMV-sr39tk (provided by Sam Gambhir, Stanford University, CA, USA). Gene targeting was performed as described[19, 48]. In brief, 60 μg of AclI-linearized pRosa26-mT/sr39tk were electroporated into R1 ES cells[49]. After 8 days of selection with 320 μg mL⁻¹ G418 (GIBCO, Life Technologies, Darmstadt, Germany), 200 clones were isolated and expanded. Fourteen correctly targeted ES cell clones carrying the R26-mT/sr39tk knock-in allele were identified by Southern blot analysis of EcoRV-digested genomic DNA with a probe that binds 5′ to the integration site[50]. R26-mT/sr39tk mice were generated by injection of ES cells into 3.5 dpc C57BL/6N blastocysts. Male chimeras were mated with C57BL/6N females to obtain heterozygous R26-mT/sr39tk mice (B6;129-Gt(ROSA)26Sor^tm2(ACTB-tdTomato,-sr39tkFeil)) on a mixed 129 Sv/C57BL6N genetic background. Germline transmission of the modified R26 allele was verified by Southern blot analysis of tail DNA. Heterozygous male R26-mT/sr39tk mice showed a severely reduced reproductive rate

(no offspring from >10 different heterozygous R26-mT/sr39tk males). Heterozygous female R26-mT/sr39tk mice were fertile and were backcrossed to C57BL/6N animals for three to eight times before experimental animals were generated.

**Genotyping**. PCR-based genotyping of ear biopsies was done for R26-mT/sr39tk mice with primers ROSA10 (5′-CTCTGCTGCCTCCTGGCTTCT-3′), ROSA11 (5′-CGAGGCGGATCACAAGCAATA-3′) and ROSA04 (5′-TCAATGGGCGGG GGTCGTT-3′)[51] (Supplementary Fig. 1a), and for R26-lacZ mice with ROSA10, ROSA11 and RF127 (5′-GCGAAGAGTTTGTCCTCAACC-3′)[10]. ROSA10 and ROSA11 amplify a 330-bp fragment of the wild-type R26 locus, ROSA10 and ROSA04 amplify a 250-bp fragment of the R26-mT/sr39tk allele, and ROSA10 and RF127 amplify a ca. 200-bp fragment of the R26-lacZ allele. Mice potentially carrying both R26-mT/sr39tk and R26-lacZ alleles were genotyped with a four-primer PCR containing ROSA10, ROSA11, ROSA4 and RF127.

Routine genotyping of CD4-Cre and Myh6-Cre lines was performed with primers Cre800 (5′-GCTGCCACGACCAAGTGACAGCAATG-3′) and Cre1200 (5′—GTAGTTATTCGGATCATCAGCTACAC-3′), which amplify a 400-bp fragment of the Cre transgene. To detect the Pf4-Cre and Myh6-CreER[T2] transgenes, we used primers EiCre1 (5′-GACAGGCAGGCCTTCTCTGAA-3′) and EiCre2 (5′-CTTCTCCACACCAGCTGTGGA-3′), which amplify a 522-bp fragment[52]. Verification of Cre lines was performed with PCR primers specific for the Cre transgene as described in the respective original publications[22–24, 30].

**Generation and analysis of ES cells with activated sr39tk expression**. R26-mT/sr39tk ES cells carrying the Cre-activatable L2 allele were electroporated with the Cre-expression plasmid pIC-Cre[53] to generate ES cells with an excised mT cassette and activated sr39tk expression (L1 allele) (Supplementary Fig. 1a). ES cell clones carrying the L1 allele were identified by a combination of Southern blot analysis, fluorescence microscopy to detect loss of mT expression and growth assays to test their sensitivity to the sr39tk suicide substrate ganciclovir (see below). Out of 96 clones analyzed, 12 showed the expected DNA fragment pattern, loss of mT fluorescence and growth inhibition in the presence of ganciclovir. For further analyses, selected R26-mT/sr39tk (+/L2) and Cre-recombined (+/L1) ES cell clones were thawed and expanded from frozen replica plates.

For western blot analysis, ES cell extracts were prepared in SDS lysis buffer (2% SDS, 50 mM Tris-Cl pH 8.0, 5 mM EDTA, 100 mM NaCl). Protein content of cell extracts was measured with the Total Protein Kit, Micro Lowry, Peterson's modification (Sigma-Aldrich, Darmstadt, Germany). Cell lysates containing 20 µg protein were subjected to SDS-PAGE and western blot analysis on polyvinylidene fluoride membranes. Membranes were stained with a polyclonal rabbit HSV1-tk antiserum (1:2,000) provided by William C. Summers (Yale University, CT, USA) and horseradish-peroxidase-coupled goat anti-rabbit secondary antibody (1:5,000, Cell Signaling, Danvers, MA, USA). To confirm equal protein loading, a second gel run with the same samples was stained with Coomassie Brilliant Blue according to standard procedures.

For cell growth assays, ES cells were seeded into six-well plates with feeder cells and incubated in ES cell medium consisting of Dulbecco's modified Eagle's medium supplemented with 20% fetal calf serum (FCS) (GIBCO), 0.1 mM 2-mercaptoethanol and 1,000 U mL$^{-1}$ leukemia-inhibitory factor (ESGRO, Millipore, Darmstadt, Germany) at 37 °C and 5% CO$_2$. After 24 h, medium was changed to ES cell medium with or without 2 µM ganciclovir (Roche, Mannheim, Germany). Medium was changed every 1–2 days. Cell growth was documented with a digital camera attached to a phase contrast microscope (Axiovert 40, Zeiss, Jena, Germany).

For analysis of [18F]FHBG uptake into ES cells in culture, cells were seeded into six-well plates on feeder cells 4 days before the experiment. On the day of the experiment, medium was exchanged with 3 mL ES cell medium containing 740 kBq mL$^{-1}$ (20 µCi mL$^{-1}$) [18F]FHBG and cells were further incubated at 37°C and 5% CO$_2$. After 30, 60 and 120 min, the supernatants were collected and cells were washed twice with 1 mL phosphate-buffered saline (PBS); pooled supernatant and PBS washes of each well represented the extracellular fraction. For each time point, three replicates were prepared. Then, cells were lysed directly in the wells with 1 mL SDS lysis buffer (2% SDS, 50 mM Tris-Cl pH 8.0, 5 mM EDTA, 100 mM NaCl) per well. Cell lysates were collected and wells were washed twice with 2 mL PBS; pooled lysate and PBS washes of each well represented the intracellular fraction. Radioactivity was measured in a γ-counter (Wallac 1470 WIZARD, Perkin Elmer, Turku, Finland) and intracellular [18F]FHBG accumulation was calculated as ratio of intracellular fraction to total radioactivity measured in the intracellular and extracellular fraction.

**Flow cytometry and platelet count**. Single cell suspensions of spleen, lymph nodes and thymus were obtained via 70-µm cell strainers in fluorescence-activated cell sorting (FACS) buffer (PBS containing 5% FCS). After lysis of red blood cells with ACK lysis buffer, cells were separated via 40-µm cell strainers and stained with flurochrome-conjugated antibodies (V500-coupled anti-CD45.2, V450-coupled anti-CD4, fluorescein isothiocyanate-coupled anti-CD8 (BD Biosciences, Heidelberg, Germany); APC-coupled anti-CD3 (Biolegend)) for 30 min at 4 °C in FACS buffer. Flow cytometry was performed using a BD LSR II Flow Cytometer (BD Biosciences) and analyzed with FlowJo software (FlowJo, Ashland, OR, USA).

Platelet counts were determined in whole blood drawn from the retroorbital plexus of isoflurane-anesthetized animals into 300 µL acid citrate-dextrose buffer. Platelet numbers were determined with an automated blood analyzer (Sysmex Se 9000, Kobe, Japan).

**Cutaneous DTHR model**. Delayed-type hypersensitivity reactions (DTHRs) were induced with 2,4,6-trinitrochlorbenzene (TNCB) as described previously[29]. Briefly, mice were sensitized by application of 80 µL of 5% TNCB dissolved in a 4:1 mixture of acetone and Miglyol 812 (SASOL, Witten, Germany) to the shaved abdomen. Seven, 10 and 12 days later, mice were challenged with 20 µL of 1% TNCB dissolved in a 1 : 9 mixture of acetone and Miglyol 812 on both sides of the left ear to elicit a TNCB-specific contact hypersensitivity reaction. The degree of inflammation was assessed through measurement of ear thickness with a digital micrometer before the first TNCB challenge, 12 h after each subsequent ear challenge, and then every 3 days.

**MI model**. MI was performed by surgical ligation of the LAD coronary artery. Briefly, mice were anesthetized with 5 mg kg$^{-1}$ midazolam, 0.5 mg kg$^{-1}$ medetomidin and 0.05 mg kg$^{-1}$ fentanyl. After oro-tracheal intubation and ventilation, the thoracic cavity was surgically exposed. For permanent ligation, the LAD was ligated with a non-resorbable 8-0 filament. I/R injury was induced by ligation of the LAD for 60 min with the aid of a polyethylene tube. Ischemia and reperfusion were confirmed by discoloration and akinesia or recoloration, respectively, of the ischemic region. Sham treatment was performed without ligation of the filament. After closing of the access site, anaesthesia was antagonized by injection of atipamezol (2.5 mg kg$^{-1}$) and flumazenil (0.5 mg kg$^{-1}$), and animals were monitored until recovery. Buprenorphine (0.05 mg kg$^{-1}$) was injected subcutaneously for analgesia.

**PET tracer synthesis**. Tracers were synthetized by the Department of Preclinical Imaging and Radiopharmacy, Werner Siemens Imaging Center Tübingen. [18F]Fluoride was produced at the PETtrace cyclotron (General Electric Healthcare, Uppsala, Sweden) using [18O]water (Rotem, Leipzig, Germany) and the $^{18}$O(p,n)$^{18}$F nuclear reaction. [18F]FHBG was synthetized according to previously described procedures[54] with some modifications. [18F]Fluoride was transferred to an automated synthesis module (TRACERlab FX-FN, GE Healthcare, Liège, Belgium) and first adsorbed on an ion exchange cartridge (SEP-PAK light, Accell Plus QMA, Waters, USA) preconditioned with 10 ml of 1 N aqueous NaHCO$_3$ and 10 mL water. [18F]Fluoride was eluted and flushed into the reaction vial with a mixture of 900 µL of acetonitrile and 100 µL of water wherein 3.5 mg (25 µmol) of K$_2$CO$_3$ and 15 mg (40 µmol) Kryptofix 2.2.2. were dissolved. The solution was dried under vacuum at 60 °C for 5 min and afterwards at 120 °C for additional 5 min. Labeling was carried out using 4 mg of the precursor Tosyl-FHBG (ABX, Radeberg, Germany) in 1 mL dimethyl sulfoxide under stirring at 120 °C within 5 min. Hydrolysis was achieved with 1 mL of 1 N HCl at 100 °C in 5 min. For neutralization of the reaction mixture, 0.4 mL of 2 N aqueous NaOH was added. Unreacted [18F]fluoride was removed by passing the reaction solution through an Al$_2$O$_3$-cartridge (SEP-PAK light, Alumina N, Waters) preconditioned with 10 mL water. Afterwards, the reactor was flushed with 0.5 mL of water, which was also passed through the Al$_2$O$_3$ cartridge. The combined fractions were injected onto the high-performance liquid chromatography (HPLC) column (Phenomenex Luna column C18/2, 250 × 10 mm; 5 µm) for separation. A mixture of ethanol/50 mM Na$_2$HPO$_4$ (5/95, v/v) was used as eluent. Retention time of the product was 18–22 min at a flow rate of 4 mL min$^{-1}$, detected by UV (254 nm) and radio detector. The pH was adjusted to 7.4–7.6 by adding 0.25 mL of 0.4 N NaH$_2$PO$_4$. Finally, the product was sterile filtered through a 0.22-µm filter. Overall synthesis time was 60 min. Volume of the product was in the range of 6–10 mL, activity of the batches was 1.9–3.9 GBq. Radiochemical purity was determined via thin layer chromatography/phosphor imager analysis and HPLC and was 82–92% with both methods. Specific activity was >100 GBq µmol$^{-1}$ at the end of synthesis. [18F]FDG was synthesized in a TRACERlab MXFDG synthesizer (GE Healthcare) using previously established methods[55] and mannose triflate (ABX) as a precursor. Quality control was performed in accordance with European pharmacopoeia quality guidelines. The radiochemical purity as determined by thin layer chromatography was >95%.

**In vivo PET imaging and ex vivo analysis of tracer uptake**. Animals were imaged using dedicated Inveon small animal PET scanners (Siemens Preclinical Solutions, Knoxville, TN, USA) yielding a spatial resolution of ≈1.3 mm in the reconstructed images[56]. For [18F]FHBG imaging, animals were anesthetized for 5–10 min with 1.5% isoflurane in pure oxygen and injected with tracer into the tail vein., and 180 min after tracer injection for conscious uptake, animals were anesthetized with 1.5% isoflurane in pure oxygen and PET acquisition was performed for 20 min. For [18F]FDG imaging, ketamine (100 mg per kg body weight) and xylaxine (10 mg per kg body weight) were used for anesthesia. Sixty minutes after tracer injection into the tail vein and unconscious uptake on a heating blanket, a 10 min PET acquisition was performed. All PET data were reconstructed with an iterative ordered-subset expectation maximization algorithm.

MR imaging was performed on dedicated 7T small animal MR tomographs (Bruker Biospin MRI and ClinScan, Bruker, Ettlingen, Germany) with fiducial

markers containing [18]F activity for subsequent image fusion. A T2-weighted three-dimensional space sequence (TE/TR 205/3000 ms, matrix of 161 × 256, slice thickness 0.22 mm) was used to obtain morphological information.

For data analysis, PET and MRI images were fused, normalized to each other and analyzed using Inveon Research Workplace software (Siemens Preclinical Solutions) or Amide[57]. To analyze tracer uptake into selected organs, three-dimensional regions of interest (ROIs) were manually defined. When available, co-registered MR images were used to define ROIs; otherwise, PET tracer uptake patterns were used for ROI definition. To analyze regional uptake into the myocardium, 17-segment PMs[58] of cardiac PET images were created using PMOD software (PMOD Technologies Ltd, Zurich, Switzerland). Uptake of [18]FHBG and [18]FDG are reported as percent injected dose (ID) per cubic centimeter (%ID/c.c.m.).

To determine [18]FHBG uptake in isolated organs and tissue slices, animals were killed and dissected tissues were subjected to biodistribution analysis or autoradiography. To analyze tracer biodistribution, organ radioactivity was measured in a γ-counter (Wallac 1470 WIZARD). The fraction of injected tracer dose per gram tissue (%ID per g) was calculated by normalizing decay-corrected counts to ID and sample wet weight. For autoradiography, organs were embedded and snap-frozen in O.C.T. TissueTek Compound (Sakura Finetek, Torrance, CA, USA). Organs were cut into 20-μm sections using a CM1850 cryostat (Leica Microsystems, Wetzlar, Germany). Dry sections were placed on a phosphor screen. After 24 h, the screen was read using a STORM Phosphor-Imager (Molecular Dynamics, Sunnyvale, CA, USA). Then, sections were stained with hematoxylin and eosin. Tissue sections from mice carrying also the R26-lacZ reporter gene were stained with X-Gal as described below. Stained sections were visually examined (Axioskop, Zeiss) or scanned with a NanoZoomer 2.0-HT C9600 (Hamamatsu Photonics, Herrsching am Ammersee, Germany). Analysis of autoradiographs and photomicrographs was performed with ImageJ (National Institute of Health, Bethesda, MD, USA).

**X-Gal staining and immunohistochemistry**. X-Gal staining of whole-mount organs was performed as described[9]. Animals were transcardially perfused with Heparin-PBS (PBS with 250 mg L$^{-1}$ heparin) followed by PBS containing 2% formaldehyde and 0.2% glutaraldehyde. After dissection, organs were fixed for 1 h in the same fixative solution at room temperature, washed twice with PBS, and incubated overnight at room temperature in X-Gal staining solution (PBS at pH 7.4 containing 2.5 mM K$_3$Fe(CN)$_6$, 2.5 mM K$_4$Fe(CN)$_6$, 2 mM MgCl$_2$, 1 mg mL$^{-1}$ X-Gal). After staining, tissues were washed with PBS and stored at 4 °C in 70% ethanol. Documentation was performed with a digital camera attached to a stereo microscope (Stemi 2000, Zeiss).

X-Gal staining of blood was performed with samples drawn from the heart into Heparin-PBS. They were mixed with an equal volume of PBS containing 2% formaldehyde and 0.2% glutaraldehyde, and fixed for 15 min at room temperature. Then, samples were centrifuged at 200 g for 5 min, washed three times with PBS and incubated overnight at room temperature in X-Gal staining solution. Stained blood cells were washed three times with PBS, mixed 1:1 with aqueous mounting medium (Aquatex, Merck, Darmstadt, Germany) and examined on a bright field microscope (Axiovert 40, Zeiss).

For X-Gal staining of tissue sections, mice were transcardially perfused with Heparin-PBS followed by PBS containing 0.2% glutaraldehyde. After dissection, organs were removed and left for another hour in PBS with 0.2% glutaraldehyde at room temperature. After three washes with PBS, organs were kept in 30% sucrose overnight at 4 °C. Organs were embedded in O.C.T. TissueTek Compound (Sakura Finetek), snap-frozen and cut at −25 °C into sections of 10 or 20 μm thickness. Sections were mounted on SuperFrost Plus glass slides (Thermo Fisher, Braunschweig, Germany), dried at room temperature, incubated for 10 min in PBS containing 0.2% glutaraldehyde and 2 mM MgCl$_2$ at 4 °C, followed by three washes with PBS containing 0.1% Triton X-100 and 2 mM MgCl$_2$ at room temperature. Staining was performed overnight in X-Gal staining solution at room temperature. After three washes with PBS containing 0.1% Triton X-100 and 2 mM MgCl$_2$, sections were incubated in PBS with 4% formaldehyde for 10 min and washed again with PBS containing 0.1% Triton X-100 and 2 mM MgCl$_2$. Then, sections were counterstained with hematoxylin/eosin or nuclear fast red, dehydrated and mounted with DePeX (Merck). Embedded sections were examined on a light microscope (Axiovert 40, Zeiss) or scanned with a NanoZoomer 2.0-HT C9600 (Hamamatsu Photonics).

Immunohistochemistry was performed on 3–5-μm-thick paraffin sections stained with antibodies against CD31 (Abcam, Cambridge, UK) or Mac3 (BD Biosciences). Immunohistochemistry was performed on an automated immunostainer (Ventana Medical Systems, Oro Valley, AZ, USA) according to the company's protocols for open procedures with slight modifications. Sections were counterstained with hematoxylin. Appropriate positive and negative controls were used to confirm specificity of staining.

**Data analysis and statistical methods**. Investigators were not blinded for genotype or treatment of the animals during experiments and data evaluation. Sample sizes were chosen as minimal requirement to faithfully detect in vivo functionality of the R26-mT/sr39tk PET reporter approach (typically three to six animals per group). Statistical analyses were performed with Microcal Origin (Pro 2016; OriginLab Corporation, Northampton, MA). One-way analysis of variance was used to compare groups as detailed in the figure legends, where $*p < 0.05$,

$**p < 0.01$ and $***p < 0.001$, respectively. Normal distribution of data was not tested. Homogeneity of variance between groups was tested using the Brown–Forsythe test.

**Data availability**. All relevant data are available upon request from the corresponding author.

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

## Acknowledgements

We thank S. Aidone, B. Birk, D. Bukala, M. Harant and N. Altmeyer for excellent technical assistance; I. Gonzalez Menendez for help with preparation of figures; M. Paolillo for reading the manuscript; A. Friebe, Stefan Offermanns and A. Waisman for providing Cre mice; S. Gambhir for the sr39tk construct; L. Luo for the pRosa26-mT/mG plasmid; W.C. Summers for the HSV1-tk antiserum; and all members of R.F.'s laboratory for critical discussion. This work was supported by the Fund of Science, Deutsche Forschungsgemeinschaft (KFO 274 projects FE 438/8-2, LA 2423/4-2 and LA 2423/7-2, SFB 685 project B6 and SFB TRR156 project 03), BMBF (grant number 0314103) and the Interdisciplinary Centre for Clinical Studies (IZKF), Core Unit PIX, University of Münster, Münster, Germany.

## Author contributions

M.T. performed and analyzed most of the experiments and generated the figures. B.F.S., S.F. and Y.L. performed and evaluated PET imaging studies together with M.T. S.F. helped with the generation of R26-mT/sr39tk mice and performed fluorescence macroscopy of organs. B.F.S. and M.K. provided the DTHR model and T-cell analysis. J.V. performed the MI model. M. Golla helped with X-Gal staining of tissues. A.V. contributed to construction of the targeting vector. U.K. and L.Q.-M. did immunohistochemistry. M.O., H.F.L. and M. Gawaz helped with platelet analysis. M.E. and G.R. performed tracer synthesis. M.T., C.M.G., F.L., M.S., M.K., B.J.P. and R.F. contributed to the design of experiments. R.F. oversaw the study. M.T. and R.F. wrote the manuscript. All authors discussed and commented on the manuscript.

## Additional information

**Competing interests:** The authos declare no competing financial interests.

