## [Peer Review file · Nature Communications]

Reviewers' comments:

Reviewer #1 (Remarks to the Author):

The manuscript titled "A novel mouse model enables Cre/lox-assisted noninvasive in vivo tracking and quantification of specific cell populations by positron emission tomography" is a well written, thorough study detailing a new approach to track different cell populations by PET non-invasively. The novelty of the approach and the attention to methodology and detail is commendable and hence this reviewer recommends that the manuscript be accepted pending minor revisions detailed below:

Line 67 'For genetically inducible'

Line 74 and 75: Explain why it is not possible with current Cre transgenic mice.

Line 89-92: Re-phrase this sentence. Not very clear.

Supplementary figure 6e and Figure 3b and 3c: The authors mention that one of the mice showed high uptake of FHBG in its left uninflamed ear in the autoradiography in figure 3C. Was this also visible in the PET images? The quantification of the PET signal in Figure 3B does not seem to reflect this observation. Please discuss.

In the discussion the authors, discuss the limitation of tracking T cells in lymph nodes due to background accumulation of tracer in bone and gastrointestinal tract. As far as imaging T cells goes, not being able to track lymph nodes which are major sites of T cell accumulation would be a disadvantage if you want to follow biology. The authors should therefore also discuss the possibility of extending this approach to using alternate PET probes to sr39TK or using alternate PET reporter gene/probe strategies.

The authors should discuss ectopic activity of CD4-Cre mice in skeletal muscle. Is this unique to CD4 promoter leakiness? Do other promoters also have similar ectopic activity? Please discuss.

Discuss pros and cons of Cre approach vs other approaches like Crispr to engineer transgenic mice.

It would also be good if the authors discuss a little about what other cell types can be tracked (based on available tissue specific Cre systems) and in which diseases context given also the limitation of FHBG's background uptake.

Reviewer #2 (Remarks to the Author):

This well written manuscript describes an excellently executed set of experiments that combines a known PET reporter imaging approach with a new transgenic mouse which expresses a floxed gene for the herpes simplex virus thymidine kinase reporter gene in all tissues. When bred with the appropriate cell specific Cre strain, it allows for non-invasive quantitative cell tracking of any cell for which a Cre exists. This is a neat and novel idea and will likely be very useful for many studies to come. While it copies the concept of other reporter genes, this new approach can be done completely noninvasively with PET imaging, but nevertheless quantitatively, in many tissues that are large enough to place a sufficient number of PET voxels.

The figures are clear and the data appear solid. I have just one minor question: Is the high magnification Xgal stain in figure 4 C flipped?

Reviewer #3 (Remarks to the Author):

In the present manuscript the authors presented a novel tool for in vivo tracing of cells combining Cre-LoxP system and PET technology using sr39tk reporter gene. They could show the use of this tool for tracing specific group of cells such as platelets, T lymphocytes, and cardiomyocytes. Moreover, they demonstrated the use of this tool for tracing cells in two disease-contexts consisting of inflammation and myocardial ischemia.

This study is of general interest and provided a new tool for in vivo phenotyping although some limitations has to be faced, however this will importantly advance the investigation of cell-fate in developmental and/or diseases related questions.

The manuscript is well writing and the figure well prepared.

However, this review feels that some few points have to be addressed before being considered for publication:

1- Minor: Expression of mT before Cre recombination (in case of inducible Cre) or in non-specific cells are not clearly shown.

2- Minor: Rosa-mT/sr39tk appears in page 1 line 97 and the explanation of "mT" is later in line 108.

3-Page 5: lines 129-133: "We performed in vivo [¹⁸F]FHBG-PET imaging studies (Fig. 2) with Cre-positive experimental mice that were expected to express sr39tk in the respective target cells (sr39tk+; genotype: Cre[tg/+],R26[sr39tk/+]). To evaluate nonspecific tracer uptake, Cre-negative control animals (sr39tk-; genotype: Cre[+/+],R26[sr39tk/+]) were analyzed in parallel."

In this paragraph I have 2 comments:

Minor: The nomenclature is confusing: Cre[+/+] should mean Cre-negative? Normally animals will be designated as Crepos/+/Creneg/- R26[sr39tk/+]

Major: a Cre positive R26[+/+] is missing, especially after the data presented in Suppl. 5 where pure WT and Cre negative are similar and only the Cre positive R26[sr39tk/+] showed an effect in cell numbers in Lymph nodes, Spleen and Thymus. It is well-described that (and depending on the mouse background) Cre expression alone can cause some alteration independent of the GOI. Therefore, the sentence in page 7: line 181, "Thus, the reduced number of T cells in CD4/sr39tk mice was apparently mainly caused by expression of sr39tk and not by expression of Cre recombinase." Is overestimated.

4-Minor: Page 8: line 233, why 1 mg Tamoxifen for 5 days and why after 4 weeks?

5-General Major: The limitation of this technique in Cre-lox-inducible models has to be clearly discussed, especially in proliferative cells. They cannot be traced when they proliferate after induction of the recombination and therefore will dilute the effect of tracing; well-designed protocols have to be providing for it.

As in the present manuscript, in the case of myocardial ischemia i.e., it remains the question open whether newly formed Mhy6-expressing cells will dilute the labelling and the reduction of ([¹⁸F]FHBG) uptake and will underestimate the rate of cells viability.

6- Suppl. Figure 3 and 4

Minor: Animals are named differently as stated in the text "mice are denoted "Cre promoter/lacZ"

mice" and in the figure they are denoted "promoter-Cre". In suppl. Figure 3 the WT bgal staining (control) is missing.

Major: A co-staining will be more suitable to claim that the bgal positive cells are of a certain nature (ie. CD31, Mac3). Better magnifications are necessary for stating the staining in the smooth muscle cells rather than endothelial of bgal. Scale bars are missing.

Figure 4:

Would be helpful to have a better resolution of pictures for visualizing the I/R and Lig FHBG uptake at least for some of the panels.

Regards,

Reviewer #1:

(1) Line 67 ‘For genetically inducible’

Done (now line 60).

(2) Line 74 and 75: Explain why it is not possible with current Cre transgenic mice.

We changed the sentence to *‘With the currently available R26 Cre reporter mouse lines, however, noninvasive quantitative detection of labeled cells in vivo at the whole-body level is not possible, because detection of the aforementioned reporter proteins relies on either ex vivo methods requiring*

tissue fixation, invasive methods with a small field of view such as intravital microscopy, or semi-quantitative noninvasive methods such as bioluminescence imaging' (line 66-71).

(3) Line 89-92: Re-phrase this sentence. Not very clear.

We tried to clarify this statement (now lines 85-92): *'However, transgenic mice with a chromosomally integrated Cre-responsive PET reporter gene have not been described to date. In such a mouse line, Cre-expressing cell populations will be labeled for PET imaging through Cre-mediated activation of reporter gene expression at the genomic level. Once reporter gene expression is activated, cells and their progeny are stably labeled, even if the cells proliferate or change their phenotype, which may lead to a loss of Cre expression. This approach would permit noninvasive long-term visualization of any given cell population for which a respective cell type-specific Cre mouse line is available.'*

(4) Supplementary figure 6e and Figure 3b and 3c: The authors mention that one of the mice showed high uptake of FHBG in its left uninflamed ear in the autoradiography in figure 3C. Was this also visible in the PET images? The quantification of the PET signal in Figure 3B does not seem to reflect this observation. Please discuss.

This is correct. One animal showed relatively increased (but not really 'high') uptake into the untreated ear. We have mentioned and discussed this observation on page 8 (line 220-224): *'Autoradiography detected elevated tracer uptake also in the non-challenged right ears of sr39tk+ mice, in particular in one of the three sr39tk+ animals analyzed (Fig. 3c). This was likely due to scratching and transfer of TNCB from the left to the right ear, thereby, inducing inflammation also in the "non-challenged" right ear.'*

The PET images shown in Fig. 3b (left) are not from this mouse. For clarification, we prepared the figure below for the reviewer. It shows the individual time courses of [¹⁸F]FHBG uptake measured with PET in three individual animals (left; we added the uptake ratio between TNCB-treated and control ear in red) in comparison to the respective ear autoradiographs taken on day 20 (right). Indeed, the mouse with elevated tracer uptake in the non-challenged right ear as revealed by autoradiography on day 20 did also show increased PET signals in the untreated ear (first animal, open circles). This PET signal was similar to the TNCB-treated ear on day 20, but much weaker than the PET signals at the peak of inflammation in the challenged left ears at day 13. In general, changes in [¹⁸F]FHBG uptake of the TNCB-treated ears over time were much higher than in untreated ears. Therefore, the comparably minor changes within the untreated ears are visible in the autoradiographs, where animals are shown individually, but not in the original Fig. 3b (right), where the average of all animals was shown.

Based on the reviewer's comment, we have now revised Fig. 3b (right) and show PET signals in individual mice/ears. Along the same line, we have also revised Supplementary Fig. 6 b-d to show PET data in organs of individual animals.

(5) In the discussion the authors, discuss the limitation of tracking T cells in lymph nodes due to background accumulation of tracer in bone and gastrointestinal tract. As far as imaging T cells goes, not being able to track lymph nodes which are major sites of T cell accumulation would be a disadvantage if you want to follow biology. The authors should therefore also discuss the possibility of extending this approach to using alternate PET probes to sr39TK or using alternate PET reporter gene/probe strategies.

We agree with this notion and have now extended the discussion (line 316-327): ‘*The current limitation in quantifying T cells in lymph nodes could be overcome by increasing the purity of the [¹⁸F]FHBG preparation, which was 82-92% in the present study, thereby reducing the amount of free [¹⁸F]fluoride, which accumulates in bone. Background radioactivity in the gastrointestinal tract could be reduced by increasing [¹⁸F]FHBG elimination via the intestine or by using alternative sr39tk substrates with faster clearance such as [¹⁸F]FEAU^{34,35}. In addition, use of thymidine kinase substrates carrying [¹²⁴I] as longer-lived PET isotope has been described^{35,36}. With [¹²⁴I]-labeled radiotracers, PET imaging can be performed much later after tracer injection than with short-lived [¹⁸F]-labeled tracers, thereby, allowing more ‘non-trapped’ tracer to be eliminated before image acquisition. Another approach to reduce background signals could be the use of alternative PET reporter genes such as the human or rat sodium iodide symporter. However, the respective tracer, iodine, may show the problem of insufficient intracellular retention due to iodine efflux³⁷.*

(6) The authors should discuss ectopic activity of CD4-Cre mice in skeletal muscle. Is this unique to CD4 promoter leakiness? Do other promoters also have similar ectopic activity? Please discuss.

We think that the weak (but significant) activity of CD4-Cre mice that was detected in skeletal muscle in the biodistribution analysis (Suppl. Fig. 2e) is not due to activity in skeletal muscle fibers, but likely caused by activity in vascular smooth muscle cells of the blood vessels, which are also the reason for ectopic activity of CD4-Cre mice in the heart. This has been stated/discussed in the manuscript (line 166-168): ‘*Ectopic activity of the CD4-Cre line in some vascular smooth muscle cells could also explain the weak but significant tracer uptake that was detected ex vivo in skeletal muscle (Supplementary Fig. 2e).*’

(7) Discuss pros and cons of Cre approach vs other approaches like Crispr to engineer transgenic mice.

Does the reviewer suggest to include a general discussion on Cre/lox vs CRISPR/Cas technology? If so, we feel this would significantly extend the manuscript, which is already close to the word limit. Also, we are not sure as to how such general discussion would relate to the PET reporter mouse model we describe in this paper. If the reviewer thinks it is absolutely required, please advise us more specifically, and we will add it.

(8) It would also be good if the authors discuss a little about what other cell types can be tracked (based on available tissue specific Cre systems) and in which diseases context given also the limitation of FHBG's background uptake.

We have added a statement about Cre mice and cell types (line 380-383): *'The binary nature of the system permits labeling of a broad spectrum of cell types by crossbreeding the sr39tk reporter line with different Cre strains, several hundreds of which are available for targeting basically any cell type of interest (<http://www.creportal.org>).'*

FHBGs background uptake and potential solutions to this limitation have been discussed (see above, point 5).

Potential applications of the PET reporter mice in disease contexts have been discussed (line 403ff): *'We foresee many applications of our sr39tk PET reporter mice in preclinical research. In combination with MRI or other PET tracers (as shown in the present study), this mouse line will improve our understanding of mammalian (patho-)biology associated with migration, accumulation, death, or survival of distinct cell populations. These processes are of fundamental importance in clinical conditions such as inflammation, diabetes, atherosclerosis, thrombosis, myocardial infarction, stroke, and tumorigenesis. Our method will also be useful to elucidate endogenous mechanisms of tissue degeneration and regeneration as well as effects of therapeutic interventions. Cells derived from sr39tk PET reporter ES cells or mice can aid the development of effective cell-based therapies, which requires monitoring of the location, distribution and long-term viability of the transplanted cells in a noninvasive manner^{3,4}. In this context, sr39tk can be used not only as a PET reporter, but also as a suicide gene enabling the elimination of therapeutic cells by ganciclovir treatment, if they are causing severe adverse effects⁴⁹.*

Reviewer #2:

(1) The figures are clear and the data appear solid. I have just one minor question: Is the high magnification Xgal stain in figure 4 C flipped?

Yes, thank you for pointing this out. Changed accordingly.

Reviewer #3:

(1) Minor: Expression of mT before Cre recombination (in case of inducible Cre) or in non-specific cells are not clearly shown.

We have extended Supplementary Fig. 1 (new panel f) with macroscopic fluorescence images of mT fluorescence in various organs of R26-mT/sr39tk mice and added the following sentence to the results section (line 116-119): *'In line with previous publications^{18,19}, which used the same targeting vector but*

different reporter genes, we observed strong and ubiquitous mT expression in organs isolated from R26-mT/sr39tk mice (Supplementary Fig. 1f).'

(2) Minor: Rosa-mT/sr39tk appears in page 1 line 97 and the explanation of “mT” is later in line 108.

Thank you for this hint, we have changed the text accordingly:

Line 93ff: *'To improve cell tracking in mammals, we generated R26 knock-in mice carrying a transgene for Cre-inducible sr39tk expression under control of the ubiquitous cytomegalovirus early enhancer/chicken β -actin/ β -globin (CAG) promoter. Because these mice express membrane-targeted tandem-dimer tomato red fluorescent protein (mT) before Cre recombination and sr39tk after Cre recombination, we named them 'R26-mT/sr39tk' mice.'*

Line 107: *'Before Cre recombination, mT is expressed from the L2 allele, where “L2” stands for “two loxP sites”.'*

(3) Page 5: lines 129-133: “We performed in vivo [18F]FHBG-PET imaging studies (Fig. 2) with Cre-positive experimental mice that were expected to express sr39tk in the respective target cells (sr39tk+; genotype: Cre[tg+],R26[sr39tk/+]). To evaluate nonspecific tracer uptake, Cre-negative control animals (sr39tk-; genotype: Cre[+/+],R26[sr39tk/+]) were analyzed in parallel.” In this paragraph I have 2 comments:

Minor: The nomenclature is confusing: Cre[+/+] should mean Cre-negative? Normally animals will be designated as Crepos+/+Creneq/- R26[sr39tk/+]

We apologize if the nomenclature we used to describe mouse genotypes caused confusion. We added several clarifications all over the manuscript and hope that with these additions it will be clear to the reader that we are using the '+' symbol to indicate respective wild type alleles:

- (1) Line 119f to: *'The general appearance and viability of R26-mT/sr39tk mice (genotype: R26[sr39tk/+], where '+' denotes the wild type allele) was normal.'*
- (2) Line 128ff to: *'We mated R26-mT/sr39tk mice with Pf4-Cre²², CD4-Cre²³, or Myh6-Cre²⁴ mice with Cre[tg+] genotype (where '+' denotes the wild type allele) to induce expression of sr39tk in platelets, T lymphocytes, or cardiomyocytes, respectively (Fig. 1a, Supplementary Table 1).'*
- (3) Line 177ff to: *'In naïve CD4/sr39tk mice (genotype: Cre[tg+],R26[sr39tk/+]), we determined T cell numbers in lymph nodes, spleen and thymus by flow cytometry. Compared to wild type mice (genotype: Cre[+/+],R26[+/+]), CD4/sr39tk mice had a smaller number of CD3⁺ lymphocytes, which was primarily due to a smaller fraction of CD4⁺ T cells (Supplementary Fig. 5b-d).'*
- (4) Legend to Figure 2 (line 855ff): *'Panels show representative [¹⁸F]FHBG-PET images of sr39tk-expressing mice (sr39tk⁺; genotype: Cre[tg+],R26[sr39tk/+]) and Cre-negative control animals (sr39tk⁻; genotype: Cre[+/+],R26[sr39tk/+]); '+' denotes the wild type allele.'*
- (5) Legend to Supplementary Figure 2: *'(a-c) Representative [¹⁸F]FHBG autoradiographs from various organs of (a) Pf4/sr39tk (Pf4-Cre), (b) CD4/sr39tk (CD4-Cre), and c) Myh6/sr39tk (Myh6-Cre) mice. sr39tk-expressing mice (sr39tk⁺; genotype: Cre[tg+],R26[sr39tk/+]) were compared to Cre-negative control mice (sr39tk⁻; genotype: Cre[+/+],R26[sr39tk/+]); '+' denotes the wild type allele.'*

Major: a Cre positive R26[+/+] is missing, especially after the data presented in Suppl. 5 where pure WT and Cre negative are similar and only the Cre positive R26[sr39tk/+] showed an effect in cell numbers in Lymph nodes, Spleen and Thymus. It is well-described that (and depending on the mouse background) Cre expression alone can cause some alteration independent of the GOI.

Therefore, the sentence in page 7: line 181, “Thus, the reduced number of T cells in CD4/sr39tk mice was apparently mainly caused by expression of sr39tk and not by expression of Cre recombinase.” Is overestimated.

We fully agree with the reviewer that Cre expression alone may cause phenotypes that are independent of the expression of Cre-responsive (reporter) genes. Indeed, to test for the effect of Cre expression alone (in the absence of sr39tk), we have analyzed in Supplementary Fig. 5b-e also CD4-Cre transgenic mice that did NOT carry the sr36tk reporter gene. In contrast to CD4-Cre/sr39tk mice (that expressed Cre and sr39tk in T cells), CD4-Cre mice (that only expressed Cre in T cells) did not show major reductions in T cell numbers. These results led us to the conclusion ‘*Thus, the reduced number of T cells in CD4/sr39tk mice was apparently mainly caused by expression of sr39tk and not by expression of Cre recombinase*’ (line 185).

To clarify the genotypes used in these experiments and to relativize our observations, we have modified lines 177ff as follows: ‘*In naïve CD4/sr39tk mice (genotype: Cre[tg+],R26[sr39tk/+]), we determined T cell numbers in lymph nodes, spleen and thymus by flow cytometry. Compared to wild type mice (genotype: Cre[+/+],R26[+/+]), CD4/sr39tk mice had a smaller number of CD3⁺ lymphocytes, which was primarily due to a smaller fraction of CD4⁺ T cells (Supplementary Fig. 5b-d). It has been reported that CD4-Cre mice have reduced T cell numbers, particularly CD4⁺ T cells in the spleen²⁷. However, the CD4-Cre mice (genotype: Cre[tg+],R26[+/+]) used in our studies showed T cell numbers similar to wild type mice, except for a slightly lower number of CD8⁺ T cells in the thymus that was statistically significant (Supplementary Fig. 5b-d). Thus, the reduced number of T cells we observed in CD4/sr39tk mice was apparently mainly caused by expression of sr39tk and not by expression of Cre recombinase.*’

(4) Minor: Page 8: line 233, why 1 mg Tamoxifen for 5 days and why after 4 weeks?

Amount and duration of Tamoxifen induction were used according to the publication describing the generation and first use of the Myh6-CreER^{T2} mouse line (Takefuji, M. et al. 2012 Circulation 126(16): 1972-1982). In this publication, loss of myocardial gene expression induced by Myh6-CreER^{T2} was analyzed 2 weeks after the last Tamoxifen treatment. However, to avoid potential side effects caused by Tamoxifen or vehicle (oil), and to reduce stress to the animals, we performed myocardial infarction 4 weeks after finishing Tamoxifen injection.

(5) General Major: The limitation of this technique in Cre-lox-inducible models has to be clearly discussed, especially in proliferative cells. They cannot be traced when they proliferate after induction of the recombination and therefore will dilute the effect of tracing; well-designed protocols have to be providing for it. As in the present manuscript, in the case of myocardial ischemia i.e., it remains the question open whether newly formed Mhy6-expressing cells will dilute the labelling and the reduction of ([18F]FHBG) uptake and will underestimate the rate of cells viability.

We understand this comment in a way that the reviewer is concerned about the fact that once-labeled cells may lose the reporter gene (or reporter gene expression) when they proliferate. If this is the reviewer’s concern, we must respectfully disagree. A central strength of our approach of Cre-mediated activation of reporter gene expression is its stability independent of cell proliferation. The key is that the reporter transgene is stably integrated into the cell’s genome and, therefore, inherited to both daughter cells upon mitotic cell division without any “dilution”, even over multiple rounds of cell division. As Cre-mediated activation of the reporter gene is based on an irreversible modification of the chromosomal DNA (i.e., excision of the mT-encoding gene cassette), it is very unlikely that cells lose reporter gene expression upon proliferation. The commonly used and well-characterized CAG promoter drives sr39tk expression after Cre-mediated activation of the reporter gene; this promoter is known for strong constitutive/cell-type independent gene expression. Therefore, it is unlikely that reporter gene expression is reduced or lost even if cells change their phenotype upon proliferation. The background behind our strategy for stable cell labeling has been mentioned several times in the manuscript. We also

hope that changes in the introduction (lines 85ff; see also comment 3 from Reviewer #1) clarify that cell labeling is not lost upon proliferation of the initially labeled cells.

The reviewer also points out that, in case of myocardial infarction, newly formed cells may dilute the labeled cells so that measurement of [¹⁸F]FHBG uptake could lead to an underestimation of myocardial viability. We agree with the reviewer. In the current study, we were not able to discern whether heart tissue was regenerated via proliferation of pre-existing cardiomyocytes (which should have been labeled) or formation of new cardiomyocytes from non-cardiomyocyte progenitor cells (which were presumably not labeled). To clarify potential limitations associated with the use of tamoxifen-inducible CreER^{T2} in general and in our myocardial infarction study, we added the following text to the discussion (line 396ff): *'If tamoxifen-inducible Cre recombinases such as CreER^{T2} are used for cell labeling, it should be considered that only cells will be labeled that expressed the recombinase during the tamoxifen pulse and that these cells might later be 'diluted' by non-labeled cells originating from a pool of CreER^{T2}-negative progenitor cells. For instance, if in our myocardial infarction model newly formed cardiomyocytes were derived from sr39tk-negative progenitor cells, then these cardiomyocytes would not take up [¹⁸F]FHBG, and cell viability would be underestimated by [¹⁸F]FHBG-PET.'*

(6) Suppl. Figure 3 and 4

Minor: Animals are named differently as stated in the text “mice are denoted “Cre promoter/lacZ” mice” and in the figure they are denoted “promoter-Cre”. In suppl. Figure 3 the WT bgal staining (control) is missing.

Please, see also point 3 (genotype nomenclature). For clarification, we changed the text in line 138ff accordingly: *'To validate results obtained with R26-mT/sr39tk PET reporter mice, we tested all Cre transgenes with the well-established R26-lacZ Cre reporter line¹⁰. LacZ-expressing mice (lacZ⁺; genotype: Cre[*tg*/+],R26[lacZ/+]) are denoted “Cre promoter/lacZ” mice. These mice were used to detect Cre-mediated activation of β-galactosidase expression by X-Gal-staining of fixed tissues. Cre-negative mice (lacZ⁻; genotype: Cre[+/+],R26[lacZ/+]) were used as negative controls for X-Gal-staining (Supplementary Fig. 3 and 4).'*

Major: A co-staining will be more suitable to claim that the bgal positive cells are of a certain nature (ie. CD31, Mac3). Better magnifications are necessary for stating the staining in the smooth muscle cells rather than endothelial of bgal. Scale bars are missing

The reviewer suggests to perform double staining of X-Gal positive cells with CD31 or Mac3 to better demonstrate the nature of the positive cells. The reviewer is correct. This would be the ideal way to demonstrate the identity of the positive cells. However, co-staining with the X-Gal stain is rather difficult due to the intensity and color of the X-Gal precipitate. Since CD31 and Mac3 antibodies are specific for endothelial cells and macrophages, respectively, we have revised Supplementary Fig. 4 as suggested by the reviewer, and include higher magnifications as insets into each picture. In these new insets, it is clearly depicted that X-Gal stains the smooth muscle cells of the blood vessels as well as macrophages, whereas CD31 stains the endothelial cells sparing the smooth muscle. In addition, we have included a higher magnification of the macrophages in the lung stained with X-Gal. As also pointed out by the reviewer, we have now added the scale bars in all microphotographs.

Figure 4: Would be helpful to have a better resolution of pictures for visualizing the I/R and Lig FHBG uptake at least for some of the panels.

To allow better evaluation of tracer uptake, we have prepared a new Supplementary Fig. 7 that shows all ‘cross-sections’ of the hearts in VLA orientation for every PET acquisition for both tracers. Images are in the resolution of the original datasets.

REVIEWERS' COMMENTS:

Reviewer #1 (Remarks to the Author):

The authors have addressed all the points raised by this reviewer. We therefore recommend that the manuscript be accepted.

Reviewer #3 (Remarks to the Author):

The authors have addressed most of my concerns and improved the manuscript accordingly.

I have a remaining comment:
(from 116067_1_rebut_0 document)

(5) General Major: The limitation of this technique in Cre-lox-inducible models has to be clearly discussed, especially ...

We understand this comment in a way that the reviewer is concerned about the fact that once-labeled cells may lose the reporter gene..."

My concern is not losing the reporter gene, which is integrated (a rather rare event). My concern is that only cells irreversibly modified upon TX induction (leading to Cre activation) will transmit this modification to their progeny and switch to sr39tk expression. Cells (having the reporter integrated) that have not been recombined, will only transmit the non-recombined allele expressing mT. These remaining "non-recombined" cells can dilute the effect when they proliferate.

It is known that TX-induction of Cre is not 100% but rather around 80%. In case of non-dividing cells like cardiomyocytes, this is not causing major problems, but it could cause dilution during infarction and for rapid dividing cells in other organs. The author may discuss how to overcome this problem for highly proliferating cells.

NCOMMS-16-29329A

Response to Reviewers:

Reviewer #3:

The authors have addressed most of my concerns and improved the manuscript accordingly.

I have a remaining comment:

(5) General Major: The limitation of this technique in Cre-lox-inducible models has to be clearly discussed, especially

We understand this comment in a way that the reviewer is concerned about the fact that once-labeled cells may lose the reporter gene...”

My concern is not losing the reporter gene, which is integrated (a rather rare event). My concern is that only cells irreversible modified upon TX induction (leading to Cre activation) will transmit this modification to their progeny and switch to sr39tk expression. Cells (having the reporter integrated) that have not been recombined, will only transmit the non-recombined allele expressing mT. These remaining “non-recombined” cells can dilute the effect when they proliferate.

It is known that TX-induction of Cre is not 100% but rather around 80%. In case of non-dividing cells like cardiomyocytes, this is not causing major problems, but it could cause dilution during infarction and for rapid dividing cells in other organs. The author may discuss how to overcome this problem for highly proliferating cells.

We thank the reviewer for this clarification. We have extended the discussion accordingly (page 14, line 522-526):

‘In general, labeling of rapidly proliferating cell populations with tamoxifen-inducible CreERT2 may be mosaic and, therefore, non-recombined cells may dilute the reporter signal when they proliferate. This problem can be addressed by using a non-inducible Cre line, for instance, the Myh6-Cre line for efficient labeling of cardiomyocytes.’